# Substantial hysteresis in emergent temperature sensitivity of global wetland CH$_4$ emissions

Kuang-Yu Chang [1✉], William J. Riley [1✉], Sara H. Knox [2], Robert B. Jackson [3,4], Gavin McNicol [3], Benjamin Poulter [5], Mika Aurela [6], Dennis Baldocchi [7], Sheel Bansal[8], Gil Bohrer [9], David I. Campbell [10], Alessandro Cescatti[11], Housen Chu[1], Kyle B. Delwiche[3], Ankur R. Desai [12], Eugenie Euskirchen [13], Thomas Friborg [14], Mathias Goeckede [15], Manuel Helbig [16,17], Kyle S. Hemes [18], Takashi Hirano[19], Hiroki Iwata [20], Minseok Kang[21], Trevor Keenan [1,7], Ken W. Krauss [22], Annalea Lohila [6,23], Ivan Mammarella [23], Bhaskar Mitra [24], Akira Miyata[25], Mats B. Nilsson [26], Asko Noormets [27], Walter C. Oechel [28], Dario Papale [29], Matthias Peichl [26], Michele L. Reba[30], Janne Rinne [31], Benjamin R. K. Runkle [32], Youngryel Ryu [33], Torsten Sachs [34], Karina V. R. Schäfer[35], Hans Peter Schmid[36], Narasinha Shurpali [37], Oliver Sonnentag [17], Angela C. I. Tang [38], Margaret S. Torn [1], Carlo Trotta [29,39], Eeva-Stiina Tuittila [40], Masahito Ueyama [41], Rodrigo Vargas [42], Timo Vesala[23,43], Lisamarie Windham-Myers [44], Zhen Zhang [45] & Donatella Zona[28,46]

Wetland methane (CH$_4$) emissions ($F_{CH_4}$) are important in global carbon budgets and climate change assessments. Currently, $F_{CH_4}$ projections rely on prescribed static temperature sensitivity that varies among biogeochemical models. Meta-analyses have proposed a consistent $F_{CH_4}$ temperature dependence across spatial scales for use in models; however, site-level studies demonstrate that $F_{CH_4}$ are often controlled by factors beyond temperature. Here, we evaluate the relationship between $F_{CH_4}$ and temperature using observations from the FLUXNET-CH$_4$ database. Measurements collected across the globe show substantial seasonal hysteresis between $F_{CH_4}$ and temperature, suggesting larger $F_{CH_4}$ sensitivity to temperature later in the frost-free season (about 77% of site-years). Results derived from a machine-learning model and several regression models highlight the importance of representing the large spatial and temporal variability within site-years and ecosystem types. Mechanistic advancements in biogeochemical model parameterization and detailed measurements in factors modulating CH$_4$ production are thus needed to improve global CH$_4$ budget assessments.

A full list of author affiliations appears at the end of the paper.

Methane ($CH_4$) is the second most important climate forcing trace gas influenced by anthropogenic activities after carbon dioxide ($CO_2$)[1–3]. Wetlands are the largest and most uncertain natural $CH_4$ source, contributing 19–33% of current global terrestrial $CH_4$ emissions ($F_{CH_4}$)[4–6]. Top-down estimates from atmospheric inversion models and bottom-up estimates from in situ measurements both indicate gradual increases in natural wetland $F_{CH_4}$ from 2000 (147–180 Tg $CH_4$ $yr^{-1}$; bottom-up vs. top-down) to 2017 (145–194 Tg $CH_4$ $yr^{-1}$), although $F_{CH_4}$ estimates from both approaches vary widely[4,6]. In addition, atmospheric $CH_4$ concentrations have rapidly increased since 2007 (+6.9 ± 2.7 ppb $CH_4$ $yr^{-1}$ for 2007–2015 vs. +0.5 ± 3.1 ppb $CH_4$ $yr^{-1}$ for 2000–2006), with increases arising from both biogenic (primarily agriculture and waste sectors) and fossil fuel-related sources[7,8]. Observed atmospheric $CH_4$ concentrations have risen consistently with RCP8.5 (Representative Concentration Pathway of 8.5 W $m^{-2}$)[9] projections since 2007, and are growing relatively faster than observed increases in $CO_2$ concentrations during the same period[8].

Wetland $F_{CH_4}$ estimates are poorly constrained due to high temporal and spatial variability[10,11], compounded by insufficient measurements of fluxes (e.g., latitudinal data bias) and predictor variables (e.g., soil temperature and moisture), knowledge gaps in $CH_4$ biogeochemistry[12], and incomplete process representation in biogeochemical models[4,5,13–15]. Several factors have been suggested to regulate wetland $F_{CH_4}$ through effects on methanogenesis (i.e., production), methanotrophy (i.e., oxidation), and $CH_4$ transport, including gross primary productivity (GPP)[16], water table depth (WTD)[17], vegetation composition[18,19], redox conditions[20], substrate quality and availability[21,22], pore water $CH_4$ solubility[23], microbial community dynamics and activity[24], and temperature[25]. At ecosystem scale, some in situ observations indicate that $F_{CH_4}$ are mainly controlled by 20–35 cm depth soil temperatures and are not sensitive to WTD variations as long as anoxic conditions exist[26–28]. Although $F_{CH_4}$ appears to be positively correlated with temperature and $CH_4$ production[24–27], how to parameterize $CH_4$ production, oxidation, and emission rates in models remain key uncertainties. Reducing the uncertainties is required to improve global $CH_4$ budget assessments and increase confidence in future climate projections, as the temperature sensitivity of $CH_4$ biogeochemistry is parameterized differently among $CH_4$ models[13,14,29]. A recent meta-analysis reported that $CH_4$ production temperature sensitivities derived from laboratory cultures are consistent with those of $F_{CH_4}$ inferred from ecosystem-scale measurements and could therefore be used as an empirical basis for $F_{CH_4}$ temperature sensitivity in models[30].

However, site-specific emergent $F_{CH_4}$ temperature dependencies inferred from different measurement periods show substantial intra-seasonal variability over the course of the year[31–33], highlighting effects from other environmental drivers. For example, intra-seasonal variability may stem from hysteretic (i.e., temporally offset) microbial and abiotic interactions[34]: higher substrate availability increases methanogen biomass and $CH_4$ production and emission later in the frost-free season[33]. Similarly, higher $F_{CH_4}$ for a given GPP later in the frost-free season has been reported, which may be caused by the time required to convert GPP to methanogenesis substrates[26]. Further, changes in WTD can regulate the emergent $F_{CH_4}$ temperature sensitivity through controls on soil redox potential[31,35–37], especially when the WTD is below the site-specific rooting depth and critical zone of $CH_4$ production[17,38,39].

Here, we evaluated observationally based emergent relationships among $F_{CH_4}$, GPP, WTD, and air ($T_{air}$) and soil ($T_{soil}$) temperatures using the global FLUXNET-$CH_4$ database[40]. We analyzed data recorded in eight ecosystem types: bog, fen, marsh, peat plateau, rice paddy, salt marsh, swamp, and wet tundra that spans 207 site-years across 48 wetland and rice paddy sites (Supplemental Fig. 1 and Supplemental Table 1). The FLUXNET-$CH_4$ database provides half-hourly ecosystem-scale eddy covariance measurements of $F_{CH_4}$ and other fluxes (e.g., $CO_2$, water vapor, and energy) measured at 83 sites across the globe[40] (including uplands, wetlands, and rice paddy sites). Apparent $F_{CH_4}$ hysteresis has been observed in response to WTD[17,31], GPP[26], $T_{air}$[33], and $T_{soil}$[27,32,33] at individual sites, but has not been synthesized across ecosystem types over distinct climate zones. Here, we analyzed intra-seasonal changes in emergent dependencies of $F_{CH_4}$ on these potential controls at each site-year. We focused on relationships of $F_{CH_4}$ with $T_{air}$ because $T_{air}$ is directly relevant to climate policy and better characterized in climate models[41]. In addition, the amount of $T_{air}$ data (207 site-years) in the FLUXNET-$CH_4$ database is about twice than that of $T_{soil}$ measured at the shallowest (0–18.3 cm; 112 site-years) and deepest (32–50 cm; 97 site-years) site-specific soil depths. We show that consistent intra-seasonal changes in emergent dependencies of $F_{CH_4}$ were derived with $T_{air}$ and $T_{soil}$ measurements at the sites where both measurements were available.

We quantified emergent $F_{CH_4}$–$T_{air}$ dependencies using a quadratic relationship (Methods; Eq. 1) fit to daily measurements reported during the frost-free season (defined by $T_{air} > 0\,°C$, Methods). This quadratic functional form was chosen because it is consistent with MacroMolecular Rate Theory[33] analyses of the temperature sensitivity of $CH_4$ production and oxidation[34] and produced reliable estimates of $F_{CH_4}$ for our study sites (Supplemental Fig. 2). For each frost-free season, seasonal $F_{CH_4}$ hysteresis was quantified as changes in emergent $F_{CH_4}$–$T_{air}$ dependencies inferred from earlier and later periods separated by the maximum seasonal $T_{air}$. We did not consider $F_{CH_4}$ outside the frost-free season, although they can be important in some high-latitude wetlands[32,42]. We used two metrics to quantify intra-seasonal changes in emergent $F_{CH_4}$–$T_{air}$ dependence: (1) Normalized area of seasonal $F_{CH_4}$ hysteresis ($H_A$; i.e., the area enclosed by emergent earlier and later period $F_{CH_4}$–$T_{air}$ relationships (Fig. 1d) normalized by maximum seasonal $F_{CH_4}$ and $T_{air}$; Methods); and (2) Mean seasonal $F_{CH_4}$ hysteresis ($H_\mu$; i.e., the difference between mean daily $F_{CH_4}$ inferred from measurements taken between later and earlier periods of the frost-free season). These two metrics are conceptually similar to those used to quantify temperature hysteresis in soil respiration[43] and soil $CO_2$ concentrations[44]. Positive and negative $H_A$ and $H_\mu$ values represent higher (e.g., Fig. 1d) and lower (e.g., Supplemental Fig. 3d) $F_{CH_4}$ later (i.e., after reaching maximum seasonal $T_{air}$) in the frost-free season, respectively.

## Results and discussion

**A case study of positive seasonal $CH_4$ emission hysteresis.** As an example of seasonal hysteresis, we examined daily estimates obtained from measurements taken at the Bibai Mire in Northern Japan (JP-BBY) where $F_{CH_4}$ is insensitive to the relatively shallow WTD from 2015 to 2017[27] (Fig. 1b, c). Although the seasonality shown in $F_{CH_4}$ appears to follow $T_{air}$ (Fig. 1a, b), a time-dependent $F_{CH_4}$-$T_{air}$ relationship varies from earlier to later parts of the frost-free season (Fig. 1d–f). Specifically, plotting daily $F_{CH_4}$ as a function of $T_{air}$ results in a counterclockwise loop from beginning to end of the frost-free season. Similar hysteretic patterns were found using $T_{soil}$ (Supplemental Fig. 4) and gap-filled $CH_4$ emissions[45] (Supplemental Fig. 5), indicating that the hysteresis is not caused by time lags between $T_{soil}$ and $T_{air}$ resulting from heat transfer into the soil[46], and is not driven by biases

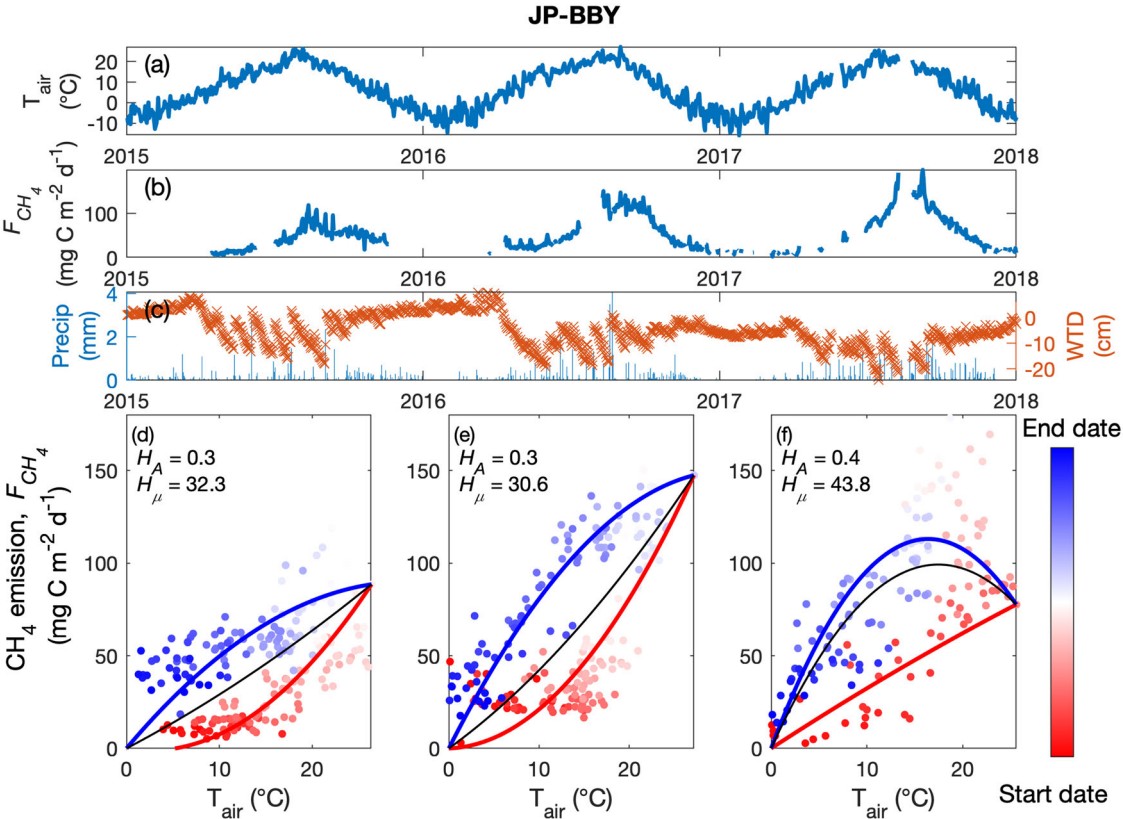

**Fig. 1 Daily mean CH$_4$ emissions have hysteretic responses to air temperature.** The quality-controlled daily air temperature (**a**), CH$_4$ emissions (**b**), precipitation (**c**, left axis), and water table depth (**c**, right axis) measured at the Bibai Mire in Japan (JP-BBY) from 2015 to 2017. CH$_4$ emission-air temperature dependencies (lines) derived from daily estimates (dots) recorded at JP-BBY for 2015 (**d**), 2016 (**e**), and 2017 (**f**). The results inferred from earlier and later parts of the frost-free season, and full frost-free season are colored in red, blue, and black, respectively. Start and end dates represent the beginning and ending of the frost-free season, respectively. Values of $H_A$ and $H_\mu$ denote the normalized area of seasonal CH$_4$ emission hysteresis (normalized area enclosed by the blue and red lines) and the mean seasonal CH$_4$ emission hysteresis calculated in each site-year, respectively.

caused by missing data. These hysteretic patterns suggest that $F_{CH_4}$ should not be represented as a single static function of $T_{air}$.

**Seasonal CH$_4$ emission hysteresis among site-years.** Overall, we detect positive seasonal $F_{CH_4}$ hysteresis in most site-years recorded in the FLUXNET-CH$_4$ database, both in terms of $H_A$ and $H_\mu$ (75–77% of site-years; Fig. 2). Consistent hysteresis patterns and magnitudes were found with monthly $F_{CH_4}$ and $T_{air}$ estimates (72–74%, Supplemental Fig. 6), indicating the observed seasonal $F_{CH_4}$ hysteresis is not sensitive to temporal resolution. The non-zero $H_A$ and $H_\mu$ values demonstrate intra-seasonal changes in emergent $F_{CH_4}$–$T_{air}$ dependencies among wetland and rice paddy sites across the globe, and their negatively skewed distribution indicates that the hysteretic responses are not likely to be random. Ignoring seasonal $F_{CH_4}$ hysteresis leads to overestimated ($28 \pm 46\%$) and underestimated ($-9 \pm 35\%$) $F_{CH_4}$ predictions earlier and later in the frost-free season across wetland and rice paddy sites, and such prediction bias is overlooked by using seasonally invariant $T_{air}$ dependence models ($-4 \pm 7\%$, Supplemental Fig. 7). For example, $F_{CH_4}$ predictions made by a seasonally invariant emergent $F_{CH_4}$–$T_{air}$ dependence at JP-BBY (i.e., black lines in Fig. 1d–f) are generally biased high and low in the earlier and later parts of the frost-free season, respectively.

To examine how potential controls are related to the observed seasonal $F_{CH_4}$ hysteresis, we analyzed the distribution pattern of $H_A$ under different site classifications and microclimatic conditions. The majority of site-years show positive seasonal $F_{CH_4}$

hysteresis when $H_A$ values are categorized into (1) different ranges of mean $T_{air}$ measured in the frost-free season (Supplemental Figs. 8), (2) different wetness conditions indicated by higher and lower mean WTD later in the frost-free season (Supplemental Fig. 9), and (3) different ecosystem types (Supplemental Fig. 10). Intra-seasonal changes in emergent GPP–$T_{air}$ dependencies show about equal site-year proportions of positive and negative $H_A$ values (48% and 52%, respectively; Supplemental Fig. 11a), suggesting that GPP does not directly contribute to the observed seasonal $F_{CH_4}$ hysteresis. Further, predominantly positive seasonal $F_{CH_4}$ hysteresis is detected using $T_{soil}$ measured at the shallowest (Supplemental Fig. 12) and deepest (Supplemental Fig. 13) site-specific soil layers, indicating substantial intra-seasonal variability in the $F_{CH_4}$-$T_{soil}$ relationship. Overall, the wetland and rice paddy observations in the current FLUXNET-CH$_4$ database suggest that $F_{CH_4}$ are generally higher later (i.e., after reaching maximum seasonal $T_{air}$ or $T_{soil}$) in the frost-free season at a given $T_{air}$ and $T_{soil}$. These hysteretic responses emerged across climate zones with various GPP and frost-free season lengths, and were not directly attributable to intra-seasonal changes in $T_{air}$ and $T_{soil}$ (Supplemental Fig. 14).

**Divergent temperature responses among sites and years.** In terms of the magnitude of seasonal $F_{CH_4}$ hysteresis, intra-seasonal changes in emergent $F_{CH_4}$-$T_{air}$ dependence vary substantially among site-years within each ecosystem type (Fig. 3), despite being predominantly positive (Fig. 2). For each ecosystem type,

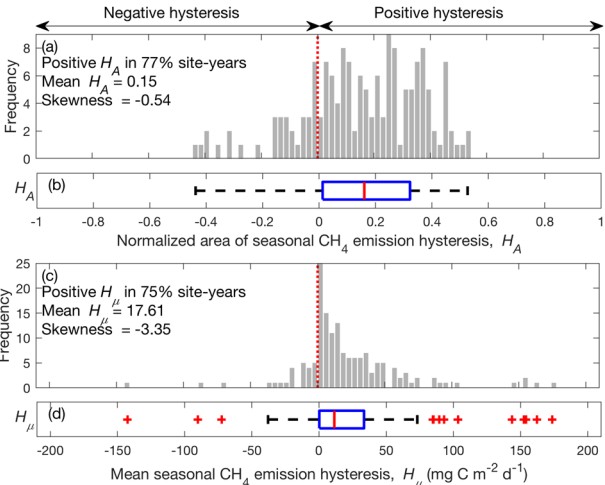

**Fig. 2 Predominantly positive seasonal CH$_4$ emission hysteresis inferred from ecosystem-scale measurements across the globe, i.e., CH$_4$ emissions are generally higher later in the frost-free season at the same temperature.** The distribution of normalized area of seasonal CH$_4$ emission hysteresis ($H_A$; **a**, **b**) and mean seasonal CH$_4$ emission hysteresis ($H_\mu$; **c**, **d**) to air temperature among site-years derived from the FLUXNET-CH$_4$ database. Positive seasonal CH$_4$ emission hysteresis indicates higher CH$_4$ emissions later in the frost-free season at the same temperature (e.g., Fig. 1d–f). Red dashed lines represent no hysteresis. The corresponding boxplot of site-year specific $H_A$ (**b**) and $H_\mu$ (**d**) derived from the FLUXNET-CH$_4$ database. The red central mark, and the bottom and top edges of the blue box indicate the median, and the 25$^{th}$ and 75$^{th}$ percentiles, respectively. The black whiskers extend to the most extreme data points not considered outliers denoted in red plus symbol.

the large inter-annual (i.e., different years within the same site) and inter-site (i.e., different site-years within the same ecosystem type) variability highlights the challenge of quantifying a universal and robust emergent $F_{CH_4}$–$T_{air}$ dependence across wetland and rice paddy sites. For example, using the Boltzmann–Arrhenius function (Methods) to represent the emergent $F_{CH_4}$-$T_{air}$ dependence of an ecosystem type cannot accurately reflect the site- and time-specific emergent relationships between $F_{CH_4}$ and $T_{air}$ (Fig. 3). A single static function of $T_{air}$ thus cannot provide accurate estimates of $F_{CH_4}$, even though meta-analyses using the same functional form suggested that such a representation would lead to consistent emergent $F_{CH_4}$–$T_{air}$ dependencies among aquatic, wetland, and rice paddy ecosystems[30]. Considering intra-seasonal variability in emergent $F_{CH_4}$–$T_{air}$ dependence leads to higher and lower apparent activation energies for $F_{CH_4}$ during earlier and later parts of the frost-free season, respectively (Supplemental Fig. 15a). Our findings indicate that the $F_{CH_4}$ temperature sensitivity is an emergent property that varies substantially with space and time and thus cannot be sufficiently generalized for formulating mechanistic CH$_4$ models, regardless of its functional form.

**Factors other than temperature modulate CH$_4$ emissions.** We applied two approaches to evaluate factors regulating the emergent $F_{CH_4}$-$T_{air}$ dependence and examine the degree of complexity needed in $F_{CH_4}$ parameterizations in biogeochemical models. In the first approach, we examined the effects of $T_{air}$, ecosystem-type variability (i.e., differences between ecosystem types), inter-site variability, inter-annual variability, and intra-seasonal variability

on $F_{CH_4}$ predictions. Specifically, $F_{CH_4}$ estimates obtained from six sets of regression models selectively representing the above-mentioned variability (Methods; Supplemental Table 2) were evaluated to investigate how spatial and temporal complexity influences model performance. In the second approach, we trained a random-forest model (Methods) with the FLUXNET-CH$_4$ database to identify factors controlling the hysteresis parameter $a_{hys}$ (Methods) that quantifies the functional relationship between $F_{CH_4}$ and $T_{air}$. To assess whether an observationally inferred model can be constructed for $F_{CH_4}$ estimates, we evaluated the predictive power of a hybrid model that uses the random-forest predicted $a_{hys}$ to describe the emergent $F_{CH_4}$–$T_{air}$ dependence (Methods; Eq. 1) in each part of the frost-free season.

The seven $T_{air}$ dependence models (six regression and one hybrid) can be broadly categorized into three tiers based on the absolute bias relative to the measured $F_{CH_4}$: (1) employing a universal emergent $F_{CH_4}$–$T_{air}$ dependence inferred from measurements across the globe without representing spatial and temporal variability (76.2% biased); (2) including ecosystem-type variability (i.e., the emergent $F_{CH_4}$–$T_{air}$ dependence is inferred from measurements collected at the same ecosystem type, so sites within an ecosystem type are uniformly represented; 63.5–63.9% biased); and (3) including ecosystem-site variability (i.e., the emergent $F_{CH_4}$–$T_{air}$ dependence is inferred from measurements collected at each site; 38.1–45.9% biased) (Fig. 4). Our results suggest that representing ecosystem-type variability does not necessarily improve $F_{CH_4}$ estimates, because the absolute bias of modeled $F_{CH_4}$ is comparable with that estimated by using a universal emergent $F_{CH_4}$–$T_{air}$ dependence, except for bog, peat plateau, and wet tundra sites (Fig. 4a). For each ecosystem type, the absolute bias of modeled $F_{CH_4}$ is reduced when ecosystem-site variability is represented, demonstrating the need to recognize inter-annual and inter-site variability (e.g., Fig. 3). For each $T_{air}$ dependence model, the absolute bias of modeled $F_{CH_4}$ is generally higher in rice paddies and salt marshes than in other ecosystem types, suggesting that $F_{CH_4}$ in these systems are sensitive to factors other than $T_{air}$. For example, timing of irrigation, drainage, planting, and harvesting can all affect $F_{CH_4}$ dynamics in rice paddies[47].

Results derived from our random-forest model confirm the importance of ecosystem-site variability in regulating $a_{hys}$ and thereby $F_{CH_4}$ predicted by the hybrid model in each part of the frost-free season (Supplemental Fig. 16). Our random-forest predictor importance analysis indicates that site-year specific $F_{CH_4}$ and $T_{air}$ values are more important for $a_{hys}$ estimates than other predictors such as latitude, GPP, and ecosystem type. The weak relationships found between seasonal $F_{CH_4}$ hysteresis and latitude (Supplemental Fig. 14h) and GPP (Supplemental Fig. 14d) are consistent with the relatively low predictor importance for $a_{hys}$ found in our random-forest model. Collectively, our results demonstrate the importance of recognizing inter-site, inter-annual, and intra-seasonal variability for the interpretation of emergent $F_{CH_4}$–$T_{air}$ dependence inferred from measurements across distinct site-years.

When using a universal emergent $F_{CH_4}$-$T_{air}$ dependence that only represents a generic $T_{air}$ sensitivity of $F_{CH_4}$ (i.e., the top row in Fig. 4a), the resulting $F_{CH_4}$ predictions substantially underestimate the range of $F_{CH_4}$ measured across wetland and rice paddy sites (Fig. 5a). This generic $T_{air}$ sensitivity of $F_{CH_4}$ flattens the high temporal and spatial variability[10,11] that strongly controls the timing and magnitude of $F_{CH_4}$, reinforcing the need

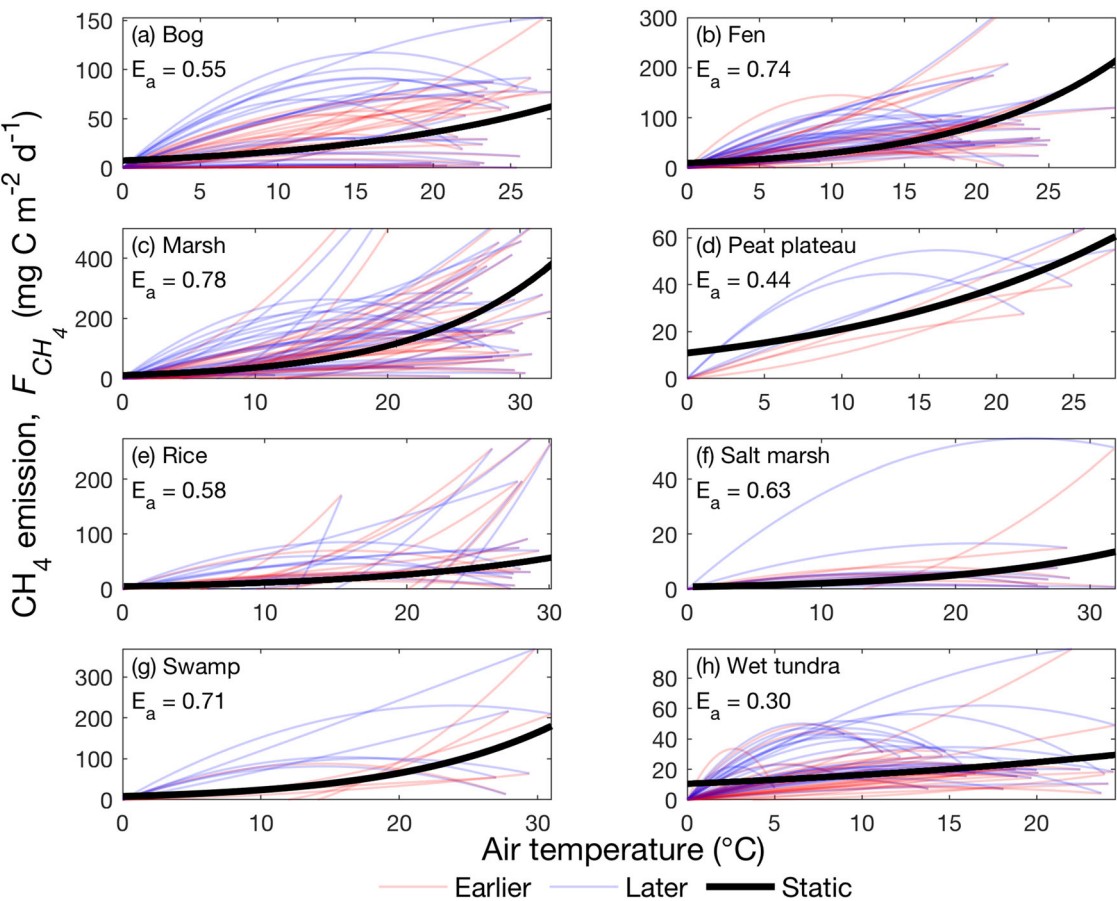

**Fig. 3 Large differences in intra-seasonal, inter-annual, and inter-site $F_{CH_4}$ emergent temperature dependencies are found for all examined ecosystem types.** Thin lines represent the site- and time-specific emergent dependencies of $CH_4$ emissions on air temperature inferred from daily measurements collected at bog (**a**), fen (**b**), marsh (**c**), peat plateau (**d**), rice paddy (**e**), salt marsh (**f**), swamp (**g**), and wet tundra (**h**) sites. Thick black lines represent ecosystem-type specific emergent dependencies of $CH_4$ emission on air temperature inferred from the Boltzmann–Arrhenius function that do not recognize spatial heterogeneity and temporal variability. The results inferred from earlier and later parts of the frost-free season, and full frost-free season are colored in red, blue, and black, respectively.

to parameterize factors other than $T_{air}$ in $CH_4$ models. Including factors other than a generic $T_{air}$ sensitivity of $F_{CH_4}$ (i.e., the bottom row in Fig. 4a) improves $F_{CH_4}$ predictions (Fig. 5b, c), which suggests that $F_{CH_4}$ and emergent $F_{CH_4}$–$T_{air}$ dependence strongly depend on site- and time-specific environmental conditions. Therefore, models should mechanistically represent $CH_4$ biogeochemistry, because site- and time-specific emergent $F_{CH_4}$-$T_{air}$ dependence cannot be accurately parameterized everywhere and all the time. Although many $CH_4$ models parameterize methanogenesis, methanotrophy, and $CH_4$ transport for $F_{CH_4}$ modeling[13], only three of 40 recently reviewed $CH_4$ models mechanistically represent $CH_4$ biogeochemistry based on explicit microbial dynamics[29]. Consequently, implementing process-based representations of $CH_4$ biogeochemistry in $CH_4$ models is necessary to improve $F_{CH_4}$ predictions across ecosystem and global scales. Such efforts are imperative because the $F_{CH_4}$ prediction error can increase substantially with increased $F_{CH_4}$, especially for the relatively simple parameterization that only represents a generic $T_{air}$ sensitivity of $F_{CH_4}$ (Fig. 5c).

**Limitations and implications**. Additional measurements and analysis of factors controlling methanogenesis, methanotrophy, and $CH_4$ transport will be needed to investigate the cause of the predominantly positive seasonal $F_{CH_4}$ hysteresis we observed

across wetland and rice paddy sites. When anoxic conditions are prevalent and $T_{soil}$ is the most important driver regulating $F_{CH_4}$[26,27] (e.g., Supplemental Fig. 4), the observed positive seasonal $F_{CH_4}$ hysteresis is consistent with the higher $F_{CH_4}$ driven by higher substrate availability later in the frost-free season[25]. We identified some environmental drivers affecting the emergent $F_{CH_4}$–$T_{air}$ dependence at sites where the necessary measurements were available: (1) When WTD drops below the critical zone of $CH_4$ production later in the frost-free season[31], the reduced FCH4 may drive negative seasonal $F_{CH_4}$ hysteresis in a given site-year (e.g., the Kopuatai bog in New Zealand (NZ-Kop), Supplemental Fig. 3). (2) $F_{CH_4}$ may become more sensitive to $T_{air}$ changes under higher salinity[48], and our results indicate that seasonal $F_{CH_4}$ hysteresis shifts from positive to negative with increased salinity (e.g., the Sacramento-San Joaquin Delta of California in USA (US-Myb), Supplemental Fig. 17).

As for the emergent $F_{CH_4}$–$T_{soil}$ dependence, our results suggest that the functional relationship between $F_{CH_4}$ and $T_{soil}$ may vary non-monotonically along the soil profile. For example, the positive seasonal $F_{CH_4}$ hysteresis inferred from $T_{soil}$ measured at 16 cm depth is stronger than those at 8 and 32 cm depths at US-Myb (Supplemental Fig. 18). Such a non-monotonic relationship indicates that the magnitude of seasonal $F_{CH_4}$ hysteresis is not

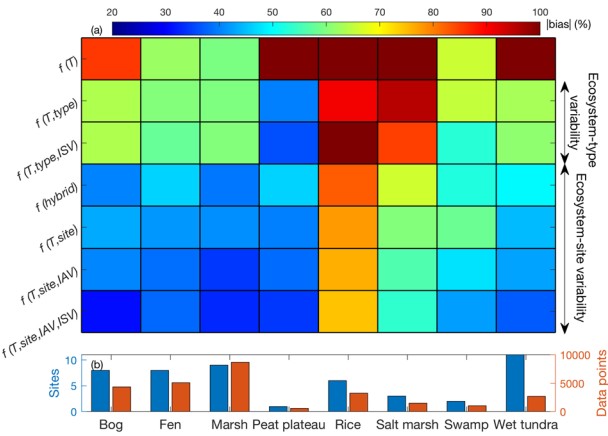

**Fig. 4 The accuracy of CH$_4$ emission estimates improves with better representation of the large wetland-site variability caused by varying environmental conditions.** The absolute bias relative to measured CH$_4$ emissions estimated by each model class for each ecosystem type (**a**). Blue and red bars denote the number of sites and quality-controlled daily data points within each ecosystem type, respectively (**b**). The abbreviations used in each model group represent air temperature (*T*), ecosystem-type variability (*type*), intra-seasonal variability (*ISV*), hybrid model based on random-forest estimated hysteresis parameter (*hybrid*), inter-site variability (*site*), and inter-annual variability (*IAV*).

simply caused by time lags between $T_{soil}$ and $T_{air}$, suggesting that factors other than temperature can strongly control $F_{CH_4}$. $T_{soil}$ measured at depths where methanogenesis is occurring will be needed to rigorously examine the emergent dependence of $F_{CH_4}$ on $T_{soil}$ across the globe, but such depth-dependent measurements are not yet available among sites in the FLUXNET-CH$_4$ database. To improve understanding of mechanisms leading to seasonal $F_{CH_4}$ hysteresis, we urge further long-term measurements on factors modulating CH$_4$ biogeochemistry (e.g., WTD, $T_{soil}$, microbial activity, and substrate availability), especially in the tropics and the Southern Hemisphere, both of which are sparsely represented in the FLUXNET-CH$_4$ database. Although seasonal $F_{CH_4}$ hysteresis occurs across seasonal climate and latitudinal gradients (Supplemental Fig. 14), better-representing ecosystems south of 30 °N could affect the partitioning of negative and positive seasonal $F_{CH_4}$ hysteresis inferred from existing measurements. While our synthesis in tropical and subtropical regions shows intra-seasonal changes in emergent $F_{CH_4}-T_{air}$ dependence (Supplemental Fig. 19), future studies are needed to examine seasonal $F_{CH_4}$ hysteresis in wetlands south of 30 °N (that account for about 75% of global wetland $F_{CH_4}$[6]).

The observed seasonal $F_{CH_4}$ hysteresis provides a benchmark to evaluate modeled $F_{CH_4}$ functional responses and should inform and motivate CH$_4$ model development and refinement. Studies have shown that temporal variations in $F_{CH_4}$ are strongly modulated by substrate and microbial dynamics[33,49,50], which may explain the substantial seasonal $F_{CH_4}$ hysteresis identified in our wetland and rice paddy sites. For example, a model that explicitly represents substrate and microbial dynamics reproduced the observed hysteretic $F_{CH_4}$ to temperature relationships in several wetlands with different vegetation and hydrological conditions[33]. Such dynamics could be parameterized in the terrestrial components of Earth system models[49]. Our synthesis thus provides observational evidence for incorporating substrate and microbial dynamics into next generation CH$_4$ models.

Using the largest available database of ecosystem-scale CH$_4$ emissions measured by eddy covariance flux towers, we show that

the apparent relationships between CH$_4$ emissions and air and soil temperatures are hysteretic and vary strongly with sampling location and measurement period. Approximately 77% of site-years recorded in the wetland and rice paddy subset of the FLUXNET-CH$_4$ database[40] show that CH$_4$ emissions become higher later in the frost-free season at the same air temperature. This predominantly positive seasonal CH$_4$ emission hysteresis may be driven by substrate-mediated higher CH$_4$ production[25] later in the frost-free season[33]. Changes in environmental conditions also modulate seasonal CH$_4$ emission hysteresis and thus ecosystem-scale CH$_4$ emissions.

Our results demonstrate that the relationship between CH$_4$ emissions and temperature is an emergent property that varies substantially across space and time. A direct integration of measurements across the globe (e.g., inferring a generic temperature sensitivity of CH$_4$ emissions) may not improve CH$_4$ model parameterization because such an approach over-simplifies factors controlling CH$_4$ emissions. Therefore, meta-analyses of CH$_4$ biogeochemistry should recognize the large intra-seasonal, inter-annual, and inter-site variability of biotic and abiotic conditions that regulate ecosystem-scale CH$_4$ emissions. Collectively, our analyses highlight the importance of observing and modeling spatial heterogeneity and temporal variability for the modeling of CH$_4$ biogeochemistry. Since most existing CH$_4$ models are developed using empirically based CH$_4$ production or emission temperature dependencies[29], our study motivates models to mechanistically represent methanogenesis, methanotrophy, and CH$_4$ transport to refine estimates of global CH$_4$ emissions and climate feedbacks[51].

## Methods

**FLUXNET-CH$_4$ database**. The FLUXNET-CH$_4$ initiative is led by the Global Carbon Project (https://www.globalcarbonproject.org) in coordination with regional flux networks (in particular AmeriFlux and the European Fluxes Database) to compile a global CH$_4$ flux database of eddy covariance and supporting measurements encompassing freshwater, coastal, natural and managed wetlands, and uplands[40]. Database descriptions, including existing sites, data standardization, gap-filling, and partitioning, have been detailed previously in Knox et al.[40]. We used daily mean temperature (air and soil), gross primary productivity as partitioned from net CO$_2$ exchange measurements, precipitation, WTD, wind speed, atmospheric pressure, and CH$_4$ emissions compiled at the 48 wetland and rice paddy sites (Supplemental Table 1) currently recorded in the FLUXNET-CH$_4$ database. Soil temperature is often measured at different depths among different sites, and only about half of the wetland sites report WTD in the current FLUXNET-CH$_4$ database[40]. We analyzed the soil temperature reported at the shallowest and deepest measured soil layers at each site to investigate their effects on regulating CH$_4$ emissions. The wetland and rice paddy data (207 site-years with 62,384 site-days as of this publication) were categorized into eight CH$_4$ emitting ecosystem types: bog, fen, marsh, peat plateau, rice paddy, salt marsh, swamp, and wet tundra, based on previous classification[52,53]. While gap-filled data are examined, they are not included in our discussion to eliminate potential biases caused by the gap-filling procedure[45].

**Frost-free season**. We define the frost-free season as the period when the observed temperature (air or soil) is >0 °C to investigate the emergent temperature responses to CH$_4$ emissions ($F_{CH_4}$) during the biologically active season across distinct climatic zones. Other data sampling thresholds, such as above-zero GPP and above 5% of annual GPP maximum, were examined, and positive seasonal $F_{CH_4}$ hysteresis is identified in 68–81% of site-years (Supplemental Figs. 20, 21), consistent with those inferred from frost-free season. We chose to present the frost-free season results because substantial GPP (e.g., above 5% of annual GPP maximum) is detected when air temperature is well below 0 °C (Supplemental Fig. 22) that may complicate our discussion of varying $F_{CH_4}$ led by temperature changes.

**Emergent temperature dependence calculation and the hysteresis parameter $a_{hys}$.** Emergent dependence of CH$_4$ emission ($F_{CH_4}$) on temperature (air or soil) is determined by fitting frost-free-season daily measurements of $F_{CH_4}$ and air and soil temperatures with a quadratic equation (Eq. 1), the Boltzmann–Arrhenius equation (Eq. 2), and first, second, third, and fifth order polynomials. Daily $F_{CH_4}$ estimates made by site- and time-specific emergent $F_{CH_4}$ temperature (air or soil) dependence models based on the above-mentioned functional forms show comparable root mean square errors (Supplemental Fig. 2). Results inferred from the

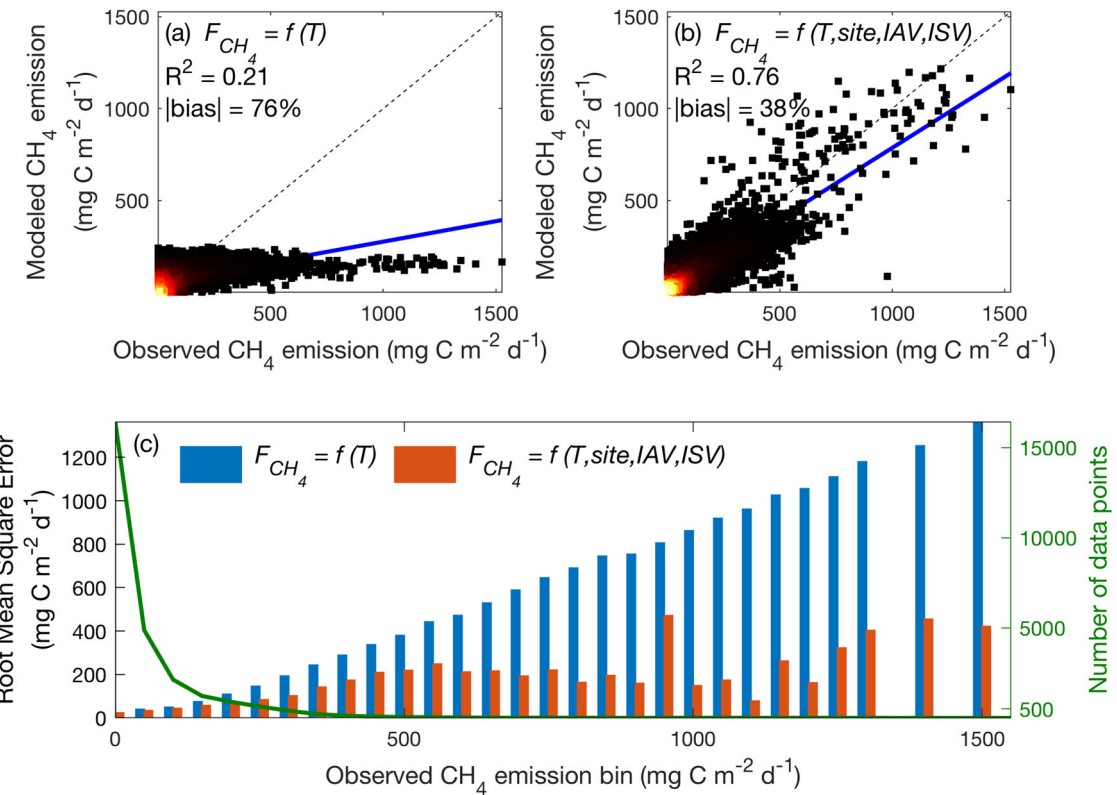

**Fig. 5 CH$_4$ emission prediction error increases substantially as measured CH$_4$ emission increases.** The performance of CH$_4$ emissions modeled by the regression models that only include a universal emergent CH$_4$ emission temperature dependence (**a**), and those that include site- and time-specific conditions (**b**). The root mean square errors associated with the regression models used in (**a**) and (**b**) (bars, left axis) and number of data points (green line, right axis) for measured CH$_4$ emission bins (**c**). Two of the 27,130 daily observations have CH$_4$ emission above 1600 mg C m$^{-2}$ d$^{-1}$, which are not shown for the ease of representation. Lighter colors in the density scatter plot represent denser data points. Solid blue and dashed black lines represent the linear best-fit and one-to-one lines, respectively. The abbreviations used in each model group represent air temperature (*T*), intra-seasonal variability (*ISV*), inter-site variability (*site*), and inter-annual variability (*IAV*).

quadratic equation (Eq. 1) are selected because (1) its functional form is mathematically consistent with the second-order polynomial equation of temperature for methanogenesis inferred from the MacroMolecular Rate Theory[54,55]; and (2) it can prescribe seasonal $F_{CH_4}$ hysteresis with a single site- and time- specific parameter ($a_{hys}$, defined below).

The fits based on the quadratic equation were forced to pass through the origin (assuming zero $F_{CH_4}$ at 0 °C, discussed below) and $F_{CH_4}$ measured at maximum seasonal temperature in each site-year using the Matlab (MathWorks Inc., 2019, version 9.7.0) polyfix function (downloaded from https://www.mathworks.com/matlabcentral/fileexchange/54207-polyfix-x-y-n-xfix-yfix-xder-dydx). The resulting emergent dependence of $F_{CH_4}$ on temperature at any given time period can thus be represented as:

$$F_{CH_4}(T) = a_{hys} \bullet T^2 + \left(\frac{F_{CH_4,T_{max}}}{T_{max}} - a_{hys} \bullet T_{max}\right) \bullet T \quad (1)$$

The symbols used in Eq. 1 denote CH$_4$ emission ($F_{CH_4}(T)$, mg C m$^{-2}$ d$^{-1}$), hysteresis parameter ($a_{hys}$, mg C m$^{-2}$ d$^{-1}$ °C$^{-2}$), daily mean temperature (*T*, °C; air or soil), maximum seasonal temperature ($T_{max}$, °C), and CH$_4$ emission measured at maximum seasonal temperature ($F_{CH_4,T_{max}}$, mg C m$^{-2}$ d$^{-1}$). Therefore, the functional relationship between and temperature, described by a quadratic equation (Eq. 1), is only determined by the value of hysteresis parameter ($a_{hys}$) and site-year variables ($F_{CH_4,T_{max}}$ and $T_{max}$).

The two constraints (passing through the origin and $F_{CH_4}$ measured at maximum seasonal temperature) imposed in Eq. 1 are intended to force the two (earlier and later part of the frost-free season) emergent $F_{CH_4}$ temperature (air or soil) dependencies to form a closed apparent hysteresis loop for each frost-free season. By doing so, seasonal $F_{CH_4}$ hysteresis can be quantified as the normalized area enclosed by the two fits, and intra-seasonal changes can be consistently compared among site-years across distinct climate zones. Ignoring $F_{CH_4}$ around 0 ° C has small effects on the magnitude and distribution of seasonal $F_{CH_4}$ hysteresis inferred from the current FLUXNET-CH$_4$ database, although substantial $F_{CH_4}$ may continue when air temperature is around or below 0 °C[32,42]. To quantify the effect of ignoring $F_{CH_4}$ around 0 °C, we replaced the constraint of zero $F_{CH_4}$ at 0 °C by the

mean $F_{CH_4}$ measured between −0.5 and 0.5 °C at 0 °C for each site-year, and found that the resulting patterns of seasonal $F_{CH_4}$ hysteresis (Supplemental Fig. 23) are consistent with those assuming zero $F_{CH_4}$ at 0 °C (Fig. 2).

**Seasonal CH$_4$ emission hysteresis.** We apply a quadratic equation (Eq. 1) to calculate the emergent dependence of CH$_4$ emission ($F_{CH_4}$) on temperature at the earlier ($F_{CH_4,earlier}(T)$) and later ($F_{CH_4,later}(T)$) part of the frost-free season separated by maximum seasonal temperature ($T_{max}$). Two metrics are used to quantify the observed seasonal $F_{CH_4}$ hysteresis: (1) Normalized area of seasonal $F_{CH_4}$ hysteresis ($H_A$), defined as the area enclosed by emergent dependencies of $F_{CH_4}$ on temperature inferred from earlier and later parts of the frost-free season (i.e.,

$H_A = \frac{\int_0^{T_{max}}(F_{CH_4,later}(T)-F_{CH_4,earlier}(T))dT}{\max(abs(F_{CH_4,earlier}(T), F_{CH_4,later}(T)))\cdot T_{max}}$); and (2) mean seasonal $F_{CH_4}$ hysteresis ($H_\mu$), defined as the difference between mean daily $F_{CH_4}$ inferred from measurements taken between later and earlier parts of the frost-free season. In each site-year, positive seasonal $F_{CH_4}$ hysteresis occurs when higher $F_{CH_4}$ are measured later in the frost-free season at a given air or soil temperature. Hysteretic patterns are similar when using either air temperatures (Fig. 1) or soil temperatures (Supplemental Fig. 4), and with either gap-filled (Supplemental Fig. 5) or non-gap-filled (Fig. 1) $F_{CH_4}$[45]. Results derived from air temperature (Fig. 2), soil temperature measured at the shallowest soil layer (Supplemental Fig. 12), and soil temperature measured at the deepest soil layer (Supplemental Fig. 13) all indicate predominantly positive seasonal $F_{CH_4}$ hysteresis across the wetland and rice paddy sites. We chose to present results derived from air temperature for its longer and more continuous record in the wetland and rice paddy subset of FLUXNET-CH$_4$ database, although soil temperature has been shown to be a better predictor for $F_{CH_4}$[33,42]. Specifically, there are 207, 112, and 97 site-years of measurements of air temperature, soil temperature measured at the sallowest soil layer (0–18.3 cm), and soil temperature measured at the sallowest soil layer (32–50 cm), respectively.

**Temperature dependence model groups.** The measurements extracted from the FLUXNET-CH$_4$ database were analyzed by seven air temperature ($T_{air}$)

dependence model groups (six regression models and a hybrid model) to evaluate factors modulating $CH_4$ emission predictions. We design the six regression models to selectively represent the effects of ecosystem-site variability and ecosystem-type variability on $CH_4$ emission prediction by labeling data points into different groups. The relationship between $CH_4$ emission and $T_{air}$ is analyzed at each part of the frost-free season, each site-year, each site, and each ecosystem type to quantify intra-seasonal, inter-annual, inter-site, and ecosystem-type variability, respectively (Supplemental Table 2). For the hybrid model, we use the hysteresis parameter predicted by our random-forest model to inform the quadratic equation (Eq. 1) for $CH_4$ emission estimates. The performance of each $T_{air}$ dependence model group was evaluated to determine the most important model components required for accurate $CH_4$ emission estimates.

**Random-forest model selection**. We used random-forest model selection to identify the most important predictors of the hysteresis parameter $a_{hys}$ (Eq. 1) that determines the functional form of emergent $CH_4$ emission air temperature dependence and thereby wetland $CH_4$ emissions ($F_{CH_4}$). Instead of $F_{CH_4}$, the hysteresis parameter $a_{hys}$ was analyzed, so the results can provide useful information on the source of observed $F_{CH_4}$ hysteresis with an understandable functional form (Eq. 1). Moreover, the most important predictors identified by the machine-learning approach can be compared with the results derived from the other approach using a range of temperature dependence model groups (Supplemental Table 2).

Ten potential predictors were selected for their relatively high predictor importance to $a_{hys}$: seasonal branch (i.e., earlier or later part in the frost-free season), GPP cumulated in a seasonal branch, precipitation cumulated in a seasonal branch, maximum seasonal temperature, mean temperature in a seasonal branch, ecosystem type, latitude, site, site-year, and $F_{CH_4}$ measured at maximum seasonal temperature. Other potential predictors, including observational year, mean WTD in a seasonal branch, mean wind speed in a seasonal branch, and mean atmospheric pressure in a seasonal branch were examined and showed limited predictive power on $a_{hys}$. Four potential predictors (seasonal branch, ecosystem type, site, and site-year) were labeled as categorical data and the rest were labeled as numerical data in our random-forest model. The random-forest model selection was performed by the Statistics and Machine-Learning Toolbox in Matlab (MathWorks Inc., 2019, version 9.7.0).

**Apparent activation energy for $CH_4$ emissions**. We quantify the apparent activation energy for $CH_4$ emissions by fitting frost-free-season daily measurements of $CH_4$ emission and air temperature with the Boltzmann–Arrhenius equation of the form:

$$\ln F_{CH_4}(T) = \bar{E}_a \bullet \left(\frac{-1}{kT}\right) + \varepsilon \qquad (2)$$

where $F_{CH_4}(T)$ is the rate of $CH_4$ emission at absolute air temperature T. $\bar{E}_a$ (in eV) and $\varepsilon$ correspond to the fitted apparent activation energy (slope) and base reaction rate (intercept), respectively. $k$ is the Boltzmann constant ($8.62 \times 10^{-5}$ eV K$^{-1}$). When the large inter-site, inter-annual, and intra-seasonal variability is muted, the apparent activation energy for $CH_4$ emission inferred from each ecosystem type is within the range reported in recent meta-analyses[30].

## Data availability
This work used publicly available FLUXNET-$CH_4$ Dataset acquired and shared by the FLUXNET community. All related data is publicly available for download at https://fluxnet.org/.

## Code availability
Code used in the analysis presented in this study is available online, and can be accessed at https://github.com/ckychang/FCH4_hysteresis[56].

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

## Acknowledgements

This research was supported by the RUBISCO SFA of the Regional and Global Modeling Analysis (RGMA) program in the Climate and Environmental Sciences Division (CESD) of the Biological and Environmental Research (BER) Program in the U.S. Department of Energy Office of Science under contract DE-AC02-05CH11231. This work was also conducted as a part of the Wetland FLUXNET Synthesis for Methane Working Group supported by the John Wesley Powell Center for Analysis and Synthesis of the U.S. Geological Survey. Funding for AmeriFlux data resources was provided by the U.S. Department of Energy's Office of Science. The compilation of the FLUXNET-CH4 data is supported by the Gordon and Betty Moore Foundation through Grant GBMF5439 "Advancing Understanding of the Global Methane Cycle" to Stanford University supporting the Methane Budget activity for the Global Carbon Project (globalcarbonproject.org). Observations at US-OWC were supported by the U.S. Department of Energy (DE-SC0021067), ODNR (Subaward N18B 315-11), and OWDA (7880). SE-Deg and SE-Sto have received funding from Swedish Research Council (ICOS-SE, grant no. 2015-06020). B.R.K.R. was supported by NSF Award 1752083. T.F.K. acknowledges support from the RUBISCO SFA, and additional support from a DOE Early Career Research Program award #DE-SC0021023. W.O. and D.Z. acknowledge support from NASA ABOVE NNX16AF94A, NSF OPP 1204263 and 1702797, EU H2020 INTAROS 629727890, and NERC UAMS NE/P002552/1. W.O. was supported by NOAA CESSRST NA16SEC4810008. N.J.S. acknowledges funding from Academy of Finland through Grant 296887. L.W.M. was supported by the LandCarbon Program of the U.S. Geological Survey. A.R.D. acknowledges DOE Ameriflux Network Management Project award to ChEAS core site cluster and NSF 0845166 for US-Los. M.U. was supported by JSPS KAKENHI (20K21849) and the Arctic Challenge for Sustainability II (ArCS II; JPMXD1420318865). E.S.T. acknowledges Academy of Finland (project codes 287039 and 330840). M.K. was supported by the National Research Foundation of Korea (NRF-2018 R1C1B6002917). C.T. thanks to the support of the E-SHAPE EU H2020 project (GA 820852). D.P. thanks the support of the DIBAF-Landscape 4.0 Departments of Excellence-2018 Program of the Italian Ministry of Research. Any use of trade, firm, or product names is for descriptive purposes only and does not imply endorsement by the U.S. Government. We acknowledge the FLUXNET-CH4 contributors for the data provided in our analyses.

## Author contributions

All authors contributed to this work. K.Y.C. and W.J.R. designed the analysis, K.Y.C. performed the analysis, and K.Y.C. and W.J.R. analyzed the results and wrote the paper, with inputs from all authors. M.A., D.B., G.B., D.I.C., A.C., A.R.D., E.E., T.F., M.G., M.H., K.S.H., T.H., H.I., M.K., K.W.K., A.L., I.M., B.M., A.M., M.B.N., A.N., W.C.O., M.P., M.L.R., J.R., B.R.K.R., Y.R., T.S., K.V.R.S., H.P.S., N.S., O.S., A.C.I.T., E.S.T., M.U., R.V., T.V., L.W.M., and D.Z. collected the observations. S.H.K., R.B.J., G.M., B.P., S.B., H.C., K.B.D., T.K., D.P., M.T., C.T., and Z.Z. build and maintain the FLUXNET-CH4 database.

## Competing interests

The authors declare no competing interests.

## Additional information

[1]Climate and Ecosystem Sciences Division, Lawrence Berkeley National Laboratory, Berkeley, CA, USA. [2]Department of Geography, The University of British Columbia, Vancouver, BC, Canada. [3]Department of Earth System Science, Stanford University, Stanford, CA, USA. [4]Woods Institute for the Environment and Precourt Institute for Energy, Stanford, CA, USA. [5]NASA Goddard Space Flight Center, Biospheric Sciences Laboratory, Greenbelt, MD, USA. [6]Finnish Meteorological Institute, Helsinki, Finland. [7]Department of Environmental Science, Policy & Management, UC Berkeley, Berkeley, CA, USA. [8]U.S. Geological Survey, Northern Prairie Wildlife Research Center, Jamestown, ND, USA. [9]Department of Civil, Environmental and Geodetic Engineering, The Ohio State University, Columbus, OH, USA. [10]School of Science, University of Waikato, Hamilton, New Zealand. [11]European Commission, Joint Research Centre (JRC), Ispra, Italy. [12]Department of Atmospheric and Oceanic Sciences, University of Wisconsin-Madison, Madison, WI, USA. [13]University of Alaska Fairbanks, Institute of Arctic Biology, Fairbanks, AK, USA. [14]Department of Geosciences and Natural Resource Management, University of Copenhagen, Copenhagen K, Denmark. [15]Max Planck Institute for Biogeochemistry, Jena, Germany. [16]School of Geography and Earth Sciences, McMaster University, Hamilton, ON, Canada. [17]Département de Géographie & Centre d'Études Nordiques, Montréal, QC, Canada. [18]Woods Institute for the Environment, Stanford University, Stanford, CA, USA. [19]Graduate School of Agriculture, Hokkaido University, Sapporo, Japan. [20]Department of Environmental Science, Faculty of Science, Shinshu University, Matsumoto, Japan. [21]National Center for AgroMeteorology, Seoul, South Korea. [22]U.S. Geological Survey, Wetland and Aquatic Research Center, Lafayette, LA, USA. [23]Institute for Atmosphere and Earth System Research/Physics, Faculty of Science, University of Helsink, Helsinki, Finland. [24]Department of Ecology and Conservation Biology, Texas A&M University, College Station, TX, USA. [25]Institute for Agro-Environmental Sciences, National Agriculture and Food Research Organization, Tsukuba, Japan. [26]Department of Forest Ecology and Management, Swedish University of Agricultural Sciences, Umeå, Sweden. [27]Department of Ecosystem Science and Management, Texas A&M University, College Station, TX, USA. [28]Department of Biology, San Diego State University, San Diego, CA, USA. [29]DIBAF, Università degli Studi della Tuscia, Largo dell'Università, Viterbo, Italy. [30]United States Department of Agriculture, Agricultural Research Service, Delta Water Management Research Service, Jonesboro, AR, USA. [31]Department of Physical Geography and Ecosystem Science, Lund University, Lund, Sweden. [32]Department of Biological and Agricultural Engineering, University of Arkansas, Fayetteville, AR, USA. [33]Department of Landscape Architecture and Rural Systems Engineering, Seoul National University, Seoul, South Korea. [34]GFZ German Research Centre for Geoscience, Potsdam, Germany. [35]Department of Biological Sciences, Rutgers University Newark, Newark, NJ, USA. [36]Institute of Meteorology and Climatology – Atmospheric Environmental Research (IMK-IFU), Karlsruhe Institute of Technology (KIT), Garmisch-Partenkirchen, Germany. [37]Production Systems, Natural Resources Institute Finland, Maaninka, Finland. [38]Sarawak Tropical Peat Research Institute, Sarawak, Malaysia. [39]Euro-Mediterranean Center on Climate Change, CMCC IAFES, Viterbo, Italy. [40]School of Forest Sciences, University of Eastern Finland, Joensuu, Finland. [41]Graduate School of Life and Environmental Sciences, Osaka Prefecture University, Osaka, Japan. [42]Department of Plant and Soil Sciences, University of Delaware, Newark, DE, USA. [43]Institute for Atmosphere and Earth System Research, Forest Sciences, Faculty of Agriculture and Forestry, University of Helsinki, Helsinki, Finland. [44]Water Mission Area, U. S. Geological Survey, Menlo Park, CA, USA. [45]Department of Geographical Sciences, University of Maryland, College Park, MD, USA. [46]Department of Animal and Plant Sciences, University of Sheffield, Sheffield, UK. ✉email: ckychang@lbl.gov; wjriley@lbl.gov

