## [Peer Review File · Nature Communications]

REVIEWER COMMENTS

Reviewer #1 (Remarks to the Author):

The authors analyse a database of CH₄ eddy covariance flux measurements collected over a range of ecosystems across the world, albeit biased towards the northern midlatitudes. They report a hysteresis of the relationship between air temperature and the CH₄ fluxes with stronger relationships at the end of the frost-free season than at the start of the frost-free season. They argue convincingly that this effect should be taken into account by models otherwise they will (as they currently do) overestimate CH₄ emissions at the start of the season and underestimate CH₄ emissions at the end of the season.

This paper certainly piqued my interest and this reviewer wholeheartedly agrees with the main message about the inadequacies of current models. But I was left disappointed after reading it for two reasons: 1) the study doesn't offer any explanation for the hysteresis and 2) the study doesn't provide any tractable way forward.

Current regional/global models are still far from representing ecosystem scales so we need to understand whether this hysteresis behavior occurs on larger spatial scales. I am sure the authors are well aware of the discrepancy between the scales captured by flux towers and those described by the current generation of models. A related comment: this reviewer appreciates why the authors chose their functional form to describe temperature dependence of CH₄ emissions but does this functional form remain valid at the larger scales described by the data? Certainly, I was not convinced by the fits shown in Figure 1 and the comparable Figures in the SI. At first mention of the random forest, I thought the authors might be developing a data-driven model - is that out of scope for this work?

The authors report that most of the sites show a positive hysteresis (Figure 2) but I am curious to see an example of negative hysteresis and to understand where they are located. This brings me to uncertainties. I didn't find any discussion of fitting uncertainties. How robust are the results? In general, I strongly suggest the authors provide some assessment of uncertainties.

The authors try to generalise some of their findings but without success. There are a lot of intra-seasonal, inter-seasonal and site-to-site variations that preclude making any useful generalities. Even the authors suggest that CH₄ emissions from rice paddies will be influenced by timings of irrigation, drainage, planting, etc. So why include these in the bulk analysis? This reviewer would go further to suggest that the short-term and long-term environmental histories of these ecosystems play a substantial role in determining CH₄ emissions at these sites, e.g. warmer early frost-free periods, atmospheric deposition, etc. I suggest the authors consider whether there is some temperature memory in these ecosystems. Will a warmer start of the season result in larger emissions later in the season? Should be easy to test given the data the authors have collated. It is difficult to eyeball that kind of effect in Figure 1 without the authors plotting temperature anomalies, for example.

Without a more robust understanding of this temperature-CH₄ emission relationship, particularly for northern midlatitudes, it is going to be tough to support tractable model development. The alternative, as suggested by the authors, is to develop process-based models of CH₄ biogeochemistry but do we have sufficient information to constrain the wide variety of required inputs, e.g. substrate availability, atmospheric deposition of labile material? I agree that if we had good quality data to support those inputs this would be the ideal solution but alas...

Reviewer #2 (Remarks to the Author):

I read the manuscript 'Substantial hysteresis in emergent temperature sensitivity of global wetland

CH4 emissions' with great interest. I was impressed by the wealth of data the authors used by the analyses they presented. I find the analyses very complete (but please see my recommendations below) and well-performed. I would also like to mention that I greatly appreciated the very clear and informative graphs. The authors managed to summarize a lot of information in easy-to-grasp figures. I believe the analyses and conclusions are novel and that the manuscript fits the scope of NComms. My two major queries refer to the relation between CH4 emission and GPP and the degree in which (local) Fch4 is underestimated when season is not taken into account. For minor suggestions and observations, please see below.

GPP- FCH4

In lines 230-233 and corresponding figures you use a rather indirect way of assessing the possible effect of GPP on FCH4. Why not use a more direct analysis here? Eg splitting your FCH4 H values in classes of positive and negative HA GPP-Tair classes or simpler: testing for a direct relation between FCH4 H values and frost free season GPP? One might expect the emergent temperature sensitivity of FCH4 to be lower at sight with low GPP as substrates for methanogenesis may become limited. In line with this, I think your statement in line 298 where you refer to a weak relationship between seasonal hysteresis FCH4 and GPP would profit from a more direct analysis.

Underestimation FCH4

In lines 217- 223 you mention that ignoring intra-seasonal variability leads to an overestimation of FCH4 early in the season and an underestimation later in the season. I recommend to include what the overall (quantitative) effect is. You could calculate this for each system or system-year and present the range possibly also include it in a supplementary figure 6c. I imagine this range would be one of the key outcomes of your analyses as it answers the overall question 'how do the current models underestimate (or overestimate in some cases?) FCH4 as determined by the seasonally-split temperature-CH4 models.

Minor issues:

Lines 158-161. Temperature influences methane solubility which in turn influences the water-atmosphere flux. Warming early in the year may therefore result in a different temperature-FCH4 relationship than cooling later in the year. In Aben et al. (2017) we found that this had only minor impact on the observed FCH4 in the experimental systems we used, but these systems had relatively shallow sediments. I wonder what the effect in the field is – where more CH4 is stored in the thicker sediment/soil/peat layer and the effect is potentially greater. I suggest to mention the change in solubility as an additional factor.

Lines 191 and methods (line 457). Change 'changes in mean...' in 'difference between mean daily FCH4'? I suspect that is what you meant, at first read it was not fully clear to me.

Line 309 some abbreviations are only explained later

Lines 308-311 It seems logical that adding these variables to the model will improve the fit. The question is how much does it improve the fit. You show this in figure 5, but it would be nice to quantitatively mention this in the text as well.

In addition to this, how much do the different variables contribute to the model fit? Maybe I overlooked the outcome of the statistical test of the models in Suppl. Table 2?

Line 335- 338 What do you mean with a strengthened coupling? More variance explained in the Tair FCH4 model? In addition, I miss a brief explanation on what the underlying mechanism could be.

Line 429 – 435 Please mention the units.

Legend figure 5 – I suggest to explain the used abbreviations again so the figure can be understood

stand alone.

Supplemental figure 6 – Why do you use bar graphs here? A boxplot would likely give more insight in the variation of your results.

I wish you good luck with the revision,

Sarian Kosten

Reference

Aben, R. C. H., N. Barros, E. van Donk, T. Frenken, S. Hilt, G. Kazanjian, L. P. M. Lamers, E. T. H. M. Peeters, J. G. M. Roelofs, L. N. de Senerpont Domis, S. Stephan, M. Velthuis, D. B. Van de Waal, M. Wik, B. F. Thornton, J. Wilkinson, T. DelSontro, and S. Kosten. 2017. Cross continental increase in methane ebullition under climate change. *Nature communications* 8:1682.

Reviewer #3 (Remarks to the Author):

The study by Chang et al. 'Substantial hysteresis in emergent temperature sensitivity of global wetland CH₄ emissions' evaluates the relationship between current static temperature dependant CH₄ flux (FCH₄) models and with other drivers using the extensive data available from the global Fluxnet database sites and proposes a more accurate approach.

Using machine-learning models, they show the importance of using site specific data and ecosystem type to account for changes in FCH₄ and air temp occurring through different period of the wetland growing seasons – shown through hysteresis trends. Data is utilised from 48 wetland/rice paddy locations from around the globe, mainly from mid-high latitude northern hemisphere stations.

3 years of 'frost free' FCH₄ data from Bibai Mire in Japan (2015-2017), along with soil and air temperature changes, WTD, GPP and precipitation, demonstrated a clear 'case study' showing the clear hysteresis pattern between FCH₄ and air (and soil) temperature. Using random-forest predictor analysis they found air temp was the most important predictor for FCH₄ and further show that their hysteresis modelling accuracy improves air temp dependency for FCH₄, and by using site specific conditions for each model.

The data utilised is extensive, accessing '207 site years' of information to explore hysteresis patterns, showing that static air temp models can over and underestimate FCH₄ predictions by 27 and -9% during different parts of the growing seasons. The manuscript is very well written, presented in a logical order and uses clear visuals, data and SI to support their results and discussion, and is an appropriate fit for the target journal.

Most of my comments are minor, however I have one major concern that requires addressing but should be easily achieved.

Although there is one sentence at line 351 admitting a lack of low latitude FLUXNET site data, this presents a major caveat in the global context of the paper. The lack of low to mid latitude, and southern hemisphere site data, absent in synthesizing the results, model and conclusions requires some more discussion and explanation as these important biogeochemical methane hotspots are always frost free, feature high GPP, high rainfall, and subsequently, high rates of methanogenesis and FCH₄. Tropical CH₄ emissions account for ~65% of the total global budget, ~30 % mid latitude and

high latitude = ~4%.

Therefore I believe the lack of data from tropical and subtropical regions - responsible for the majority of global wetland methane emissions - requires some further consideration, commentary and discussion within the manuscript. For example:

Does the predominantly high latitude data used for the model truly represent and apply to modeling 'global wetland methane emissions' - as the title of paper suggests? As far as I can tell, 45 of the 48 sites have latitudes >30 degrees away from the equator and only 1 site within in the southern hemisphere.

What would the authors expect if their analysis was created with more tropical system data and subsequent FCH₄ models (~65% of the global methane budget)? Or subtropical (~30% GMB) and southern hemisphere wetlands that are generally frost-free sites?

Although somewhat speculative due to low n, could the authors infer and present a short term tropical case study analysis using data from the FLUXNET tropical station (MY-MLM, Malaysia) in the manuscript? And a low-mid latitude subtropical case study?

At these sites, the range in T_{air} and T_{soil} seasonally may be smaller, and the growing season much longer. Therefore would the authors expect a less pronounced hysteresis pattern? Would this mean to the FCH₄ results? Would other drivers such as monsoonal wetness vs dryness become more important predictors than temperature?

Without some clarification surrounding the implications of the analysis, modelling and results without tropical wetland locations (and lesser extent southern hemisphere sites) woven into the manuscript, I don't feel that use of 'global wetland emissions' and extrapolations 'across the globe' are necessarily accurate or globally representative, in the current version of the manuscript.

Further, if the model was used by others to calculate tropical FCH₄ and future climate scenarios (i.e high fluxing tropical zones - which as the authors show are potentially more sensitive to simplistic T_{air} models - Fig. 5C), would this future work over or underestimate FCH₄ using this new model? How could this error or bias be reduced?

By addressing some of these points and suggesting ways forward (and/or adding some caution) when using the new model, will hopefully aid others to reduce any unintended and inherent global FCH₄ bias.

Minor comments as follows:

Line 120-125 - citation #4 and these values stated here are not the 'current global terrestrial emissions' and require updating. See Saunio et al., (2020) for the current global CH₄ budget: Saunio, et al. (2020). The global methane budget 2000-2017. *Earth System Science Data*, 12(3), 1561-1623. These and reference through should be updated inline with the 2000-2017 global methane budget.

Line 134 - please briefly state which 'predictor variables'?

Also, latitudinal data bias, differences in methodological approach are also compounding factors adding to uncertainty of FCH₄. Also under 'knowledge gaps', wetland forests represent a globally overlooked CH₄ source (see Barba et al (2019), Methane emissions from tree stems: a new frontier in the global carbon cycle. *New Phytol*, 222: 18-28. doi:10.1111/nph.15582)

Line 138 - what about oxygen availability and other terminal electron acceptors/substrate availability? Similar to knowledge gaps above, under 'vegetation composition' as a regulator of global FCH₄ - trees

account for 50% of Amazon FCH₄ – 'closing the Amazon methane budget' so could be cited here. Pangala, et al. (2017). Large emissions from floodplain trees close the Amazon methane budget. *Nature*, 552(7684), 230-234.

Line 164 – This wording is slightly misleading as reads as though the data from 83 sites was used, therefore and I suggest you just write something along the lines of: ...(utilising the data from 48 wetland and rice paddy sites, of the 83 available in the FLUXNET-CH₄ network).

Line 184 – how is oxidation utilised in this theory and model?

Line 320 - As FCH₄ is highest in tropical wetlands, and as the authors state, can increase prediction error with simple Tair parameterizations, could the data from the MY-MLM swamp be used to 'calibrate' and test this for a tropical wetland, to address my comments above?

End of review comments.

We thank the reviewers and editor for their careful review of our paper, and believe these suggestions have allowed us to improve the paper. Our responses (in blue) to the reviewers' comments (**in bold**) are below:

Referee #1 comments:

The authors analyse a database of CH₄ eddy covariance flux measurements collected over a range of ecosystems across the world, albeit biased towards the northern midlatitudes. They report a hysteresis of the relationship between air temperature and the CH₄ fluxes with stronger relationships at the end of the frost-free season than at the start of the frost-free season. They argue convincingly that this effect should be taken into account by models otherwise they will (as they currently do) overestimate CH₄ emissions at the start of the season and underestimate CH₄ emissions at the end of the season.

The authors thank the Reviewer for the valuable comments and suggestions to strengthen the analysis presented in our manuscript. We address the specific comments below.

This paper certainly piqued my interest and this reviewer wholeheartedly agrees with the main message about the inadequacies of current models. But I was left disappointed after reading it for two reasons: 1) the study doesn't offer any explanation for the hysteresis and 2) the study doesn't provide any tractable way forward.

Regarding the reviewer's first point, we were not able to mechanistically explain the seasonal methane emission (F_{CH_4}) hysteresis observed at our sites because we do not have sufficient comprehensive measurements to estimate belowground energy, water, and carbon exchanges that determine CH₄ production, oxidation, and transport. However, we (and others) have published results explaining hysteresis at individual sites. Therefore, in the revised manuscript we discuss factors modulating the relationship between F_{CH_4} , temperature and substrate dynamics (Lines 154-164; Lines 373-378). This revised text is based on our recently published papers (Chang et al. 2020; Mitra et al. 2020) and discussions among our large group of co-authors. In Chang et al. (2020), we found that substrate availability modulates methanogen biomass and activity, which leads to higher F_{CH_4} later in the season due to substrate accumulation. Several of our co-authors have published papers concluding that hysteretic F_{CH_4} is due to changes in substrate availability and water table depth. Other authors have proposed relationships between GPP, NPP, substrate availability, and microbial activity.

Regarding the reviewer's second point about tractable ways forward, our conclusions regarding processes that need to be represented in next generation CH₄ models are now clarified in the revised manuscript (Lines 371-381). We hope the newly included discussion can guide future research in this field and provide a tractable way forward.

The added discussion is as follows: "The observed seasonal F_{CH_4} hysteresis provides a novel benchmark to evaluate modeled F_{CH_4} functional responses and should inform and motivate CH₄ model development and refinement. Studies have shown that temporal variations in F_{CH_4} are strongly modulated by substrate and microbial dynamics^{33,52,53}, which may explain the substantial seasonal F_{CH_4} hysteresis identified in our wetland and rice paddy sites. For example, a model that explicitly represents substrate and microbial dynamics reproduced the observed hysteretic F_{CH_4} to temperature relationships in several wetlands with different vegetation and hydrological conditions³³. Such dynamics could be parameterized in the terrestrial components of Earth system models⁵². Our synthesis thus provides observational evidence for incorporating substrate and microbial dynamics into next generation CH₄ models."

Current regional/global models are still far from representing ecosystem scales so we need to understand whether this hysteresis behavior occurs on larger spatial scales. I am sure the authors are well aware of the discrepancy between the scales captured by flux towers and those described by the current generation of models. A related comment: this reviewer appreciates why the authors chose their functional form to describe temperature dependence of CH₄ emissions but does this functional form remain valid at the larger scales described by the data? Certainly, I was not convinced by the fits shown in Figure 1 and the comparable Figures in the SI. At first mention of the random forest, I thought the authors might be developing a data-driven model - is that out of scope for this work?

We agree with the reviewer that emergent F_{CH_4} temperature dependence likely depends on spatial scale. However, large-scale land surface models are often calibrated and benchmarked by flux tower measurements before applying them to regional and global scales (e.g., Collier et al. 2018). Therefore, the F_{CH_4} hysteresis observed in our flux tower measurements highlights the need to improve CH₄ biogeochemistry representations in the current generation of CH₄ models, as simulation errors related to inaccurate temperature sensitivity are likely to propagate from ecosystem scales to larger spatial scales.

The performance of our temperature dependence function (based on the MMRT approach (Schipper et al. 2014)) is comparable to the Arrhenius function widely used in biogeochemical and Earth system models, as we now show in the revised manuscript (Supplemental Fig. 2). In addition, we used two metrics to quantify F_{CH_4} hysteresis in our manuscript: (1) normalized area of seasonal F_{CH_4} hysteresis (H_A) and (2) mean seasonal F_{CH_4} hysteresis (H_μ). Both H_A and H_μ indicate intra-seasonal variations in emergent F_{CH_4} temperature dependence, and only H_A is calculated from the MMRT functional form.

Importantly, we only use the temperature dependence function to quantify the magnitude of F_{CH_4} hysteresis and do not propose using it in CH_4 model development. We clarified this point in the revised manuscript (Lines 267-269). We also include in the revised manuscript the distribution of apparent activation energies for F_{CH_4} inferred from measurements collected from the earlier, later, and full period of the frost-free season (Supplemental Fig. 15). This analysis resulted in comparable intra-seasonal variations in emergent F_{CH_4} temperature dependence as the other two metrics. We have to impose a temperature functional form because the available measurements are insufficient to build an accurate data-driven model that replicates F_{CH_4} hysteresis at individual sites, likely due to the large site level variability shown in our analysis. Although development of such a model is an intriguing idea, it is outside the scope of this paper.

The authors report that most of the sites show a positive hysteresis (Figure 2) but I am curious to see an example of negative hysteresis and to understand where they are located. This brings me to uncertainties. I didn't find any discussion of fitting uncertainties. How robust are the results? In general, I strongly suggest the authors provide some assessment of uncertainties.

We discuss a case study for negative F_{CH_4} hysteresis at (1) the Kopuatai bog in New Zealand, where declines in water table lead to lower F_{CH_4} later in the frost-free season (lines 343-345; Supplemental Fig. 3); and (2) the Sacramento-San Joaquin Delta of California in USA as salinity increases (lines 345-349; Supplemental Fig. 17). In the revised manuscript, we show that negative F_{CH_4} hysteresis is observed globally across latitudes (Supplemental Fig. 14h), suggesting that F_{CH_4} hysteresis is likely modulated by microclimatic conditions rather than geographical location. As described in our response to your previous comment, we evaluated the performance of our temperature dependence function and compared it to other approaches (Supplemental Fig. 2). Further, we also inferred consistent F_{CH_4} temperature hysteresis from intra-seasonal variations (1) with and without low temperature adjustment (Fig. 2a and Supplemental

Fig. 23), (2) using temporal mean F_{CH_4} (Fig. 2b), and (3) using an apparent activation energy for F_{CH_4} (Supplemental Fig. 15). We thank the reviewer for suggesting this additional analysis since the results buttress our conclusions regarding the consistency of intra-seasonal F_{CH_4} temperature hysteresis.

The authors try to generalise some of their findings but without success. There are a lot of intra-seasonal, inter-seasonal and site-to-site variations that preclude making any useful generalities. Even the authors suggest that CH₄ emissions from rice paddies will be influenced by timings of irrigation, drainage, planting, etc. So why include these in the bulk analysis? This reviewer would go further to suggest that the short-term and long-term environmental histories of these ecosystems play a substantial role in determining CH₄ emissions at these sites, e.g. warmer early frost-free periods, atmospheric deposition, etc. I suggest the authors consider whether there is some temperature memory in these ecosystems. Will a warmer start of the season result in larger emissions later in the season? Should be easy to test given the data the authors have collated. It is difficult to eyeball that kind of effect in Figure 1 without the authors plotting temperature anomalies, for example.

As suggested by the reviewer, we examined the effects of temperature memory and other factors on F_{CH_4} and found weak correlation between intra-seasonal changes in temperature and seasonal F_{CH_4} hysteresis (now discussed in Lines 248-250; Supplemental Fig. 14e, f). The large intra-seasonal, inter-seasonal, and site-to-site variations suggest that static temperature dependencies should not be used as an empirical basis for CH₄ biogeochemistry parameterization. Further, future CH₄ model evaluation approaches should consider applying functional relationships like the seasonal F_{CH_4} hysteresis we demonstrate here. We have improved our writing to make this point clearer in the revised manuscript (Lines 371 – 381).

Without a more robust understanding of this temperature-CH₄ emission relationship, particularly for northern midlatitudes, it is going to be tough to support tractable model development. The alternative, as suggested by the authors, is to develop process-based models of CH₄ biogeochemistry but do we have sufficient information to constrain the wide variety of required inputs, e.g. substrate availability, atmospheric deposition of labile material? I agree that if we had good quality data to support those inputs this would be the ideal solution but alas...

We agree with the reviewer that process-based models of CH₄ biogeochemistry

are complex and require a lot of forcing information, some of which is very difficult to obtain. However, we believe it is important for the CH₄ observational and modeling communities to be aware of the observed seasonal F_{CH_4} hysteresis, and hope to motivate new model structures that are consistent with these results. We conclude that current research in this field oversimplifies the relationship between F_{CH_4} and temperature (Lines 402 – 405).

References:

- Chadburn, S. E., Aalto, T., Aurela, M., Baldocchi, D., Biasi, C., Boike, J., et al. (2020). Modeled microbial dynamics explain the apparent temperature sensitivity of wetland methane emissions. *Global Biogeochemical Cycles*, 34, e2020GB006678. <https://doi.org/10.1029/2020GB006678>.
- Chang, K.-Y., Riley, W. J., Crill, P. M., Grant, R. F., and Saleska, S. R.: Hysteretic temperature sensitivity of wetland CH₄ fluxes explained by substrate availability and microbial activity, *Biogeosciences*, 17, 5849–5860, <https://doi.org/10.5194/bg-17-5849-2020>, 2020.
- Collier, N., Hoffman, F. M., Lawrence, D. M., Keppel-Aleks, G., Koven, C. D., Riley, W. J., et al. (2018). The International Land Model Benchmarking (ILAMB) system: Design, theory, and implementation. *Journal of Advances in Modeling Earth Systems*, 10, 2731–2754. <https://doi.org/10.1029/2018MS001354>
- Mitra, B., Minick, K., Miao, G., Domec, J. C., Prajapati, P., McNulty, S. G., et al. (2020). Spectral evidence for substrate availability rather than environmental control of methane emissions from a coastal forested wetland. *Agricultural and Forest Meteorology*, 291(July 2019), 108062. <https://doi.org/10.1016/j.agrformet.2020.108062>
- Schipper, L.A., Hobbs, J.K., Rutledge, S. and Arcus, V.L. (2014), Thermodynamic theory explains the temperature optima of soil microbial processes and high Q₁₀ values at low temperatures. *Glob Change Biol*, 20: 3578-3586. <https://doi.org/10.1111/gcb.12596>

Referee #2 comments (in bold):

I read the manuscript ‘Substantial hysteresis in emergent temperature sensitivity of global wetland CH₄ emissions’ with great interest. I was impressed by the wealth of data the authors used by the analyses they presented. I find the analyses very complete (but please see my recommendations below) and well-performed. I would also like to mention that I greatly appreciated the very clear and informative graphs. The authors managed to summarize a lot of information in easy-to-grasp figures. I believe the analyses and conclusions are novel and that the manuscript fits the scope of NComms. My two major queries refer to the relation between CH₄ emission and GPP and the degree in which (local) F_{CH₄} is underestimated when season is not taken into account. For minor suggestions and observations, please see below.

Thank you for taking the time to review our manuscript, and for the helpful suggestions. In the revised manuscript, we have improved our discussion on the relationship between CH₄ emissions (F_{CH_4}) and GPP, and compared F_{CH_4} estimated with and without representing intra-seasonal variability. For our point-to-point response to reviews, please see below.

GPP- FCH₄

In lines 230-233 and corresponding figures you use a rather indirect way of assessing the possible effect of GPP on FCH₄. Why not use a more direct analysis here? Eg splitting your FCH₄ H values in classes of positive and negative HA GPP-T_{air} classes or simpler: testing for a direct relation between FCH₄ H values and frost free season GPP? One might expect the emergent temperature sensitivity of FCH₄ to be lower at sites with low GPP as substrates for methanogenesis may become limited. In line with this, I think your statement in line 298 where you refer to a weak relationship between seasonal hysteresis FCH₄ and GPP would profit from a more direct analysis.

Thank you for the helpful suggestions. We have examined the relationships between (1) seasonal F_{CH_4} hysteresis and frost-free season GPP and (2) apparent activation energy for F_{CH_4} and frost-free season GPP, and found weak correlations between them (Supplemental Fig. 14d; Supplemental Fig. 15b, c, d). We compared the distribution patterns of seasonal F_{CH_4} hysteresis and seasonal GPP hysteresis to demonstrate that the seasonal F_{CH_4} hysteresis proposed in this study is unlikely caused by the time lags between GPP and F_{CH_4} (e.g., (Mitra et al., 2020; Rinne et al., 2018)). We have revised our manuscript to reflect these changes (lines 248-250), as follows: “These hysteretic responses emerged across climate zones with various GPP and frost-free season lengths, and were not directly attributable to intra-seasonal changes in T_{air} and T_{soil} (Supplemental Fig. 14).”

Underestimation FCH₄

In lines 217- 223 you mention that ignoring intra-seasonal variability leads to an overestimation of FCH₄ early in the season and an underestimation later in the

season. I recommend to include what the overall (quantitative) effect is. You could calculate this for each system or system-year and present the range possibly also include it in a supplementary figure 6c. I imagine this range would be one of the key outcomes of your analyses as it answers the overall question ‘how do the current models underestimate (or overestimate in some cases?) FCH4 as determined by the seasonally-split temperature-CH4 models.

To address this comment, we have included the bias of F_{CH_4} predicted by F_{CH_4} -temperature relations inferred from measurements collected from different parts of the frost-free season (Lines 225-229; Supplemental Fig. 7). Ignoring intra-seasonal variations in emergent F_{CH_4} temperature dependence overestimates F_{CH_4} ($28\pm 46\%$) early in the season and underestimates F_{CH_4} ($-9\pm 35\%$) later in the frost-free season, and such prediction bias is overlooked by using seasonally invariant temperature dependence models ($-4\pm 7\%$) due to compensating errors. Therefore, models should consider factors modulating emergent F_{CH_4} temperature dependence to represent the observed F_{CH_4} hysteresis and reduce F_{CH_4} prediction uncertainty under projected climate changes.

Minor issues:

Lines 158-161. Temperature influences methane solubility which in turn influences the water-atmosphere flux. Warming early in the year may therefore result in a different temperature-FCH4 relationship than cooling later in the year. In Aben et al. (2017) we found that this had only minor impact on the observed FCH4 in the experimental systems we used, but these systems had relatively shallow sediments. I wonder what the effect in the field is – where more CH4 is stored in the thicker sediment/soil/peat layer and the effect is potentially greater. I suggest to mention the change in solubility as an additional factor.

We have included intra-seasonal changes in CH4 solubility as a potential factor modulating emergent F_{CH_4} temperature dependence in the revised manuscript (line 141).

Lines 191 and methods (line 457). Change ‘changes in mean...’ in ‘difference between mean daily FCH4’? I suspect that is what you meant, at first read it was not fully clear to me.

We have implemented the wording suggested by the reviewer in the revised manuscript (lines 196-197 and 486).

Line 309 some abbreviations are only explained later

We have revised our wording in these sentences (lines 316, 321).

Lines 308-311 It seems logical that adding these variables to the model will improve the fit. The question is how much does it improve the fit. You show this in figure 5, but

it would be nice to quantitatively mention this in the text as well. In addition to this, how much do the different variables contribute to the model fit? Maybe I overlooked the outcome of the statistical test of the models in Suppl. Table 2?

To address this comment we have included the absolute bias values for each model group in the revised manuscript (lines 288, 291, 293). We note that the most important variable is the inclusion of ecosystem-site variability, as we describe in the revised manuscript (Lines 297-299), although representing inter-annual and intra-seasonal variability can further improve model performance.

Line 335- 338 What do you mean with a strengthened coupling? More variance explained in the Tair FCH4 model? In addition, I miss a brief explanation on what the underlying mechanism could be.

To clarify, we did not examine the effect of salinity on seasonal F_{CH_4} hysteresis in this study. We have improved our discussion on how salinity may influence F_{CH_4} (lines 345-349) based on the conclusions presented in Chamberlain et al. (2019).

Line 429 – 435 Please mention the units.

We have included the units for these metrics in the revised manuscript (lines 459-462).

Legend figure 5 – I suggest to explain the used abbreviations again so the figure can be understood stand alone.

We have included the meaning of abbreviations used in this figure in the revised manuscript (lines 799-801).

Supplemental figure 6 – Why do you use bar graphs here? A boxplot would likely give more insight in the variation of your results.

We have (1) replaced the bar graphs with boxplots and (2) included results inferred from full-season in the revised manuscript, as suggested by the reviewer (Supplemental Fig. 7).

I wish you good luck with the revision,

Sarian Kosten

We thank the reviewer for the positive review, and hope our revision addresses the reviewer's concern.

References:

- Chamberlain, S. D., Hemes, K. S., Eichelmann, E., Szutu, D. J., Verfaillie, J. G., & Baldocchi, D. D. (2019). Effect of Drought-Induced Salinization on Wetland Methane Emissions, Gross Ecosystem Productivity, and Their Interactions. *Ecosystems*. <https://doi.org/10.1007/s10021-019-00430-5>
- Mitra, B., Minick, K., Miao, G., Domec, J. C., Prajapati, P., McNulty, S. G., et al. (2020). Spectral evidence for substrate availability rather than environmental control of methane emissions from a coastal forested wetland. *Agricultural and Forest Meteorology*, 291(July 2019), 108062. <https://doi.org/10.1016/j.agrformet.2020.108062>
- Rinne, J., Tuittila, E. S., Peltola, O., Li, X., Raivonen, M., Alekseychik, P., et al. (2018). Temporal Variation of Ecosystem Scale Methane Emission From a Boreal Fen in Relation to Temperature, Water Table Position, and Carbon Dioxide Fluxes. *Global Biogeochemical Cycles*, 32(7), 1087–1106. <https://doi.org/10.1029/2017GB005747>

Response by the authors (in blue) to Anonymous Referee #3 comments (in bold):

The study by Chang et al. 'Substantial hysteresis in emergent temperature sensitivity of global wetland CH₄ emissions' evaluates the relationship between current static temperature dependant CH₄ flux (FCH₄) models and with other drivers using the extensive data available from the global Fluxnet database sites and proposes a more accurate approach.

Using machine-learning models, they show the importance of using site specific data and ecosystem type to account for changes in FCH₄ and air temp occurring through different period of the wetland growing seasons – shown through hysteresis trends. Data is utilised from 48 wetland/rice paddy locations from around the globe, mainly from mid-high latitude northern hemisphere stations.

3 years of 'frost free' FCH₄ data from Bibai Mire in Japan (2015-2017), along with soil and air temperature changes, WTD, GPP and precipitation, demonstrated a clear 'case study' showing the clear hysteresis pattern between FCH₄ and air (and soil) temperature. Using random-forest predictor analysis they found air temp was the most important predictor for FCH₄ and further show that their hysteresis modelling accuracy improves air temp dependency for FCH₄, and by using site specific conditions for each model.

The data utilised is extensive, accessing '207 site years' of information to explore hysteresis patterns, showing that static air temp models can over and underestimate FCH₄ predictions by 27 and -9% during different parts of the growing seasons. The manuscript is very well written, presented in a logical order and uses clear visuals, data and SI to support their results and discussion, and is an appropriate fit for the target journal.

Most of my comments are minor, however I have one major concern that requires addressing but should be easily achieved.

Although there is one sentence at line 351 admitting a lack of low latitude FLUXNET site data, this presents a major caveat in the global context of the paper. The lack of low to mid latitude, and southern hemisphere site data, absent in synthesizing the results, model and conclusions requires some more discussion and explanation as these important biogeochemical methane hotspots are always frost free, feature high GPP, high rainfall, and subsequently, high rates of methanogenesis and FCH₄. Tropical CH₄ emissions account for ~65% of the total global budget, ~30 % mid latitude and high latitude = ~4%.

Therefore I believe the lack of data from tropical and subtropical regions - responsible

for the majority of global wetland methane emissions - requires some further consideration, commentary and discussion within the manuscript. For example:

Does the predominantly high latitude data used for the model truly represent and apply to modeling 'global wetland methane emissions' - as the title of paper suggests? As far as I can tell, 45 of the 48 sites have latitudes >30 degrees away from the equator and only 1 site within in the southern hemisphere.

What would the authors expect if their analysis was created with more tropical system data and subsequent FCH₄ models (~65% of the global methane budget)? Or subtropical (~30% GMB) and southern hemisphere wetlands that are generally frost-free sites?

Although somewhat speculative due to low n, could the authors infer and present a short term tropical case study analysis using data from the FLUXNET tropical station (MY-MLM, Malaysia) in the manuscript? And a low-mid latitude subtropical case study?

At these sites, the range in T_{air} and T_{soil} seasonally may be smaller, and the growing season much longer. Therefore would the authors expect a less pronounced hysteresis pattern? Would this mean to the FCH₄ results? Would other drivers such as monsoonal wetness vs dryness become more important predictors than temperature?

We thank the reviewer for the valuable comments and we have carefully revised our manuscript based on your suggestions. For example, we have evaluated the relationship between mean climate conditions and seasonal CH₄ emission (F_{CH_4}) hysteresis, and found none of the examined factors (e.g., seasonal temperature, GPP, frost-free season length, and latitude) has strong correlation with seasonal F_{CH_4} hysteresis (Supplemental Fig. 14). These points have been added to the revised manuscript (Lines 248-250), as follows: "These hysteretic responses emerged across climate zones with various GPP and frost-free season lengths, and were not directly attributable to intra-seasonal changes in T_{air} and T_{soil} (Supplemental Fig. 14).".

As suggested by the reviewer, we also added case studies for the lower latitude sites (MY-MLM, US-LA1, and US-LA2; Supplemental Fig. 19); we found that intra-seasonal changes in emergent F_{CH_4} temperature dependence are consistent with the results presented in the main text. Our results thus suggest that seasonal F_{CH_4} hysteresis is likely driven by site- and time- specific thermal and hydrological conditions instead of rather static ecosystem types and climate zones. Therefore, we expect to detect intra-seasonal changes in emergent F_{CH_4} temperature dependence at various regions and climate zones, although the partitioning of site-years showing positive or negative seasonal F_{CH_4} hysteresis may vary. For example, seasonal F_{CH_4} hysteresis could become primarily negative if water table depth drops below the critical zone of CH₄ production later in the frost-free season (as discussed for the Kopuatai bog site in Lines 343-345). Additional measurements reflecting intra-seasonal changes in soil thermal

and hydrological conditions would be necessary to evaluate their effects on modulating emergent F_{CH_4} temperature dependence. The revised manuscript discusses these points (Lines 359-370). Even though factors contributing to the observed seasonal F_{CH_4} hysteresis requires further research, our results demonstrate the uncertainty of ignoring the large intra-seasonal variability in emergent F_{CH_4} temperature dependence to build or evaluate CH_4 models.

The added discussion is as follows: “To improve understanding of mechanisms leading to seasonal F_{CH_4} hysteresis, we urge further long-term measurements on factors modulating CH_4 biogeochemistry (e.g., WTD, T_{soil} , microbial activity, and substrate availability), especially in the tropics and the Southern Hemisphere, both of which are sparsely represented in the FLUXNET- CH_4 database. Although seasonal F_{CH_4} hysteresis occurs across seasonal climate and latitudinal gradients (Supplemental Fig. 14), better representing ecosystems south of 30°N could affect the partitioning of negative and positive seasonal F_{CH_4} hysteresis inferred from existing measurements. While our synthesis in tropical and subtropical regions shows intra-seasonal changes in emergent F_{CH_4} - T_{air} dependence (Supplemental Fig. 19), future studies are needed to examine seasonal F_{CH_4} hysteresis in wetlands south of 30°N (that account for about 75% of global wetland F_{CH_4} ⁶).”

Without some clarification surrounding the implications of the analysis, modelling and results without tropical wetland locations (and lesser extent southern hemisphere sites) woven into the manuscript, I don't feel that use of 'global wetland emissions' and extrapolations 'across the globe' are necessarily accurate or globally representative, in the current version of the manuscript.

In the revised manuscript, we acknowledge that the lack of tropical and subtropical sites may affect the relative abundance of site-years with positive seasonal F_{CH_4} hysteresis (lines 359-370). We believe that measurements collected from tropical and subtropical regions could shed light on mechanisms leading to seasonal F_{CH_4} hysteresis, as signals other than temperature could stand out due to less pronounced seasonal cycles in temperature. We would like to keep the word “global” in our manuscript title because our data covers various microclimates across the globe and the seasonal F_{CH_4} hysteresis presented in our study is not sensitive to changes in mean climate conditions (Supplemental Fig. 14, and as described above). We could change the word “global” to “cross continental” or other terms, if the editor and reviewer believe there is a better way to describe the large number of ecosystem-scale measurements collected from different parts of the world.

Further, if the model was used by others to calculate tropical FCH4 and future climate scenarios (i.e high fluxing tropical zones - which as the authors show are potentially more sensitive to simplistic Tair models - Fig. 5C), would this future work over or underestimate FCH4 using this new model? How could this error or bias be reduced?

To clarify, we do not advocate that others should use the results shown in Fig. 5c for prediction; this point is clarified in the revised manuscript (Lines 323-325). Those results are presented to show that using a generic temperature sensitivity that only represents temperature effects is unlikely to accurately estimate F_{CH_4} , demonstrating the uncertainty of using a fixed temperature relation to parameterized F_{CH_4} in models. Importantly, whether a CH₄ model over- or under- estimates F_{CH_4} depends on how the model represents soil thermal and moisture conditions, carbon and nutrient cycling, and microbial dynamics (if applicable) at a given site, in addition to the temperature sensitivity of CH₄ emissions or production via methanogenesis. Our results demonstrate the importance of representing and evaluating emergent F_{CH_4} functional relationships for future CH₄ model evaluation and to indicate needed mechanistic representations. As discussed in our revised manuscript (lines 371-381), we recommend that models represent the relevant microbial dynamics to reduce simulation errors and biases.

By addressing some of these points and suggesting ways forward (and/or adding some caution) when using the new model, will hopefully aid others to reduce any unintended and inherent global FCH4 bias.

Thank you for your comments. We hope our revisions discussed above have addressed your concerns.

Minor comments as follows:

Line 120-125 – citation #4 and these values stated here are not the ‘current global terrestrial emissions’ and require updating. See Saunois et al., (2020) for the current global CH4 budget:

Saunois, et al. (2020). The global methane budget 2000–2017. Earth System Science Data, 12(3), 1561-1623. These and reference through should be updated inline with the 2000-2017 global methane budget.

We have updated the values with those reported in Saunois et al. (2020) (lines 121-125).

Line 134 – please briefly state which ‘predictor variables’?

Also, latitudinal data bias, differences in methodological approach are also compounding factors adding to uncertainty of FCH4. Also under ‘knowledge gaps’, wetland forests represent a globally overlooked CH4 source (see Barba et al (2019), Methane emissions from tree stems: a new frontier in the global carbon cycle. New Phytol, 222: 18-28. doi:10.1111/nph.15582)

We have added the compounding factors and knowledge gaps suggested by the reviewer to this sentence (lines 134-136).

Line 138 – what about oxygen availability and other terminal electron acceptors/substrate availability?

Similar to knowledge gaps above, under ‘vegetation composition’ as a regulator of global FCH₄ – trees account for 50% of Amazon FCH₄ – ‘closing the Amazon methane budget’ so could be cited here. Pangala, et al. (2017). Large emissions from floodplain trees close the Amazon methane budget. *Nature*, 552(7684), 230-234.

To address this comment, we have cited Pangala, et al. (2017) and included effects of redox conditions in this discussion in the revised manuscript (line 140).

Line 164 –This wording is slightly misleading as reads as though the data from 83 sites was used, therefore and I suggest you just write something along the lines of: ... (utilising the data from 48 wetland and rice paddy sites, of the 83 available in the FLUXNET-CH₄ network).

To address this concern, in the revised manuscript we specify that 48 wetland and rice paddy sites are used in this study and the FLUXNET-CH₄ includes measurements across 83 sites (lines 169, 172).

Line 184 – how is oxidation utilised in this theory and model?

The MacroMolecular Rate Theory (MMRT; Schipper et al. 2014) has been applied to explain the temperature sensitivity of CH₄ oxidation and production, but it is not a mechanistic model. Specifically, MMRT predicts the temperature sensitivity of biological processes (i.e., not specifically for CH₄ biogeochemistry), and accounts for the decline in reaction rates above a temperature optimum for enzymes and microbial growth (unlike the exponential increases prescribed by the Arrhenius equation). We did not separately assess a temperature sensitivity for individual processes contributing to F_{CH_4} (i.e., CH₄ production, oxidation, and transport), but rather applied the MMRT quadratic functional form derived by Liang et al. (2018) to quantify emergent F_{CH_4} temperature dependence inferred from the FLUXNET-CH₄ dataset.

Line 320 - As FCH₄ is highest in tropical wetlands, and as the authors state, can increase prediction error with simple Tair parameterizations, could the data from the MY-MLM swamp be used to 'calibrate' and test this for a tropical wetland, to address my comments above?

We have added a case study to examine intra-seasonal variations in emergent F_{CH_4} temperature dependence inferred from tropical and subtropical sites (Supplemental Fig. 19). Measurements collected from the MY-MLM swamp show negative seasonal F_{CH_4} hysteresis potentially due to water table deepening later in the season. Increased measurements in the tropics would certainly help to calibrate CH₄ biogeochemistry parameterization. We note that ecosystem-scale F_{CH_4} is not necessarily highest in

tropical wetlands (e.g., the relatively low F_{CH_4} measured in MY-MLM).

References:

- Liang, L. L., Arcus, V. L., Heskell, M. A., O'Sullivan, O. S., Weerasinghe, L. K., Creek, D., et al. (2018). Macromolecular rate theory (MMRT) provides a thermodynamics rationale to underpin the convergent temperature response in plant leaf respiration. *Global Change Biology*, 24(4), 1538–1547. <https://doi.org/10.1111/gcb.13936>
- Mitra, B., Minick, K., Miao, G., Domec, J. C., Prajapati, P., McNulty, S. G., et al. (2020). Spectral evidence for substrate availability rather than environmental control of methane emissions from a coastal forested wetland. *Agricultural and Forest Meteorology*, 291(July 2019), 108062. <https://doi.org/10.1016/j.agrformet.2020.108062>
- Saunois, M., Stavert, A. R., Poulter, B., Bousquet, P., Canadell, J. G., Jackson, R. B., et al. (2020). The Global Methane Budget 2000–2017. *Earth System Science Data*, 12(3), 1561–1623. <https://doi.org/10.5194/essd-12-1561-2020>
- Schipper, L.A., Hobbs, J.K., Rutledge, S. and Arcus, V.L. (2014), Thermodynamic theory explains the temperature optima of soil microbial processes and high Q10 values at low temperatures. *Glob Change Biol*, 20: 3578-3586. <https://doi.org/10.1111/gcb.12596>

REVIEWERS' COMMENTS

Reviewer #3 (Remarks to the Author):

I feel the authors have done an excellent job in addressing all of the comments and concerns I originally raised, specifically my major comment about a lack of low latitude data and southern hemisphere sites. I am now comfortable with the use of the word 'global' in the title and have no further suggestions. Once published, I feel this will be a valuable contribution to the GHG modelling literature.